# Analysis of Genetic Diversity and Population Structure of Endemic Endangered Goose (*Anser cygnoides*) Breeds Based on Mitochondrial *CYTB*

**DOI:** 10.3390/ani14101480

**Published:** 2024-05-16

**Authors:** Shangzong Qi, Suyu Fan, Haoyu Li, Yufan He, Yang Zhang, Wenming Zhao, Qi Xu, Guohong Chen

**Affiliations:** 1Key Laboratory for Evaluation and Utilization of Poultry Genetic Resources, Ministry of Agriculture and Rural Affairs, Yangzhou University, Yangzhou 225009, China; mz120221524@stu.yzu.edu.cn (S.Q.); fansuyu20030729@163.com (S.F.); m19962648891@163.com (H.L.); he15978335879@163.com (Y.H.); wmzhao@yzu.edu.cn (W.Z.); xuqi@yzu.edu.cn (Q.X.); ghchen2019@yzu.edu.cn (G.C.); 2Key Laboratory for Evaluation and Utilization of Livestock and Poultry Resources (Poultry), Ministry of Agriculture and Rural Affairs, Yangzhou 225009, China; 3Joint International Research Laboratory of Agriculture and Agri-Product Safety, Ministry of Education of China, Yangzhou University, Yangzhou 225009, China

**Keywords:** endemic endangered goose, *CYTB* region, genetic diversity, haploid layout, population expansion

## Abstract

**Simple Summary:**

The diversity and specificity of the genetic structures of rare and endemic endangered goose (*Anser cygnoides*) breeds continue to be significant focal points of global concern. In this study, the mitochondrial genome was sequenced from six endemic endangered goose breeds to compare and assess the genetic structures of the rare bird breeds (♂:♀ = 1:1). The research results show that there are 81 haplotypes among the six endemic endangered goose breeds in China. At the same time, a historical dynamic analysis of the migration of endemic endangered goose breeds was conducted. The Wright’s fixation index of endemic endangered goose breeds in China is significantly lower than that of European goose breeds. This indicates that the adaptability of endangered breeds in China is relatively low, which can easily affect the long-term survival and reproductive ability of the population. The conservation and exploitation of waterfowl genetic resources leverage the intra- and inter-population variations to enhance the traits relevant to human interests. Hence, there is an urgent need to strategically plan and safeguard the genetic resources of Chinese goose breeds while also devising suitable protection strategies.

**Abstract:**

The analysis of the genetic diversity and historical dynamics of endemic endangered goose breeds structure has attracted great interest. Although various aspects of the goose breed structure have been elucidated, there is still insufficient research on the genetic basis of endemic endangered Chinese goose breeds. In this study, we collected blood samples from Lingxiang White (LX), Yan (YE), Yangjiang (YJ), Wuzong (WZ), Xupu (XP), and Baizi (BZ) geese (*Anser cygnoides*) and used Sanger sequencing to determine the partial sequence of the cytochrome b (*CYTB*) gene in a total of 180 geese. A total of 117 polymorphic sites were detected in the 707 bp sequence of the mtDNA *CYTB* gene after shearing and correction, accounting for approximately 16.55% of the entire sequence. The AT content (51.03%) of the processed sequence was slightly higher than the GC content (48.97%), indicating a preference for purine bases. The YJ, YE, and WZ breeds had the highest population genetic diversity, with a haplotype diversity greater than 0.9 (*Hd* > 0.9) and average population nucleotide difference of 8.01 (*K* > 8.01). A total of 81 haplotypes were detected and divided into six major branches. Among the six goose breeds, there were frequent genetic exchanges among LX, YJ, YE, and WZ geese (*Nm* > 15.00). We analyzed the distribution of base-mismatch differences in goose breeds and tested their historical dynamics for neutrality in Tajima’s D and Fu’s Fs. For YJ and WZ geese, Tajima’s D > 0, but the difference was not significant (*p* > 0.05). The actual values for the two breeds exhibited multimodal Poisson distributions. The population patterns of the WZ and YJ geese are purportedly relatively stable, and the breeds have not experienced population expansions or bottleneck effects, which is consistent with the neutrality test results. This study provides new insights into the diverse genetic origins and historical dynamics that sustain endemic endangered goose breeds.

## 1. Introduction

Animal husbandry is the cornerstone of modern agriculture. The excavation, protection, and evaluation of germplasm resources are the prerequisites for solving the “stuck neck” problem at the source of China’s livestock industry. Breed resources have developed diverse productivity types and a rich genetic diversity during evolution [1]. China is rich in waterfowl. However, with the continuous expansion of human activities and increasing environmental pressures, waterfowl breeds are facing serious threats and declines [2,3]. The current “List of Reusable Breeds of Animal Genetic Resources in China” includes 30 local goose breeds. Thirteen goose breeds are close to endangered status, fourteen local breeds are on the brink of extinction, and three breeds have already become extinct, including *Caohai*, *Wenshan*, and *Simao* geese [4,5]. To protect the diversity of rare and endemic endangered goose breeds, a national resource census survey found that some local goose breeds, such as the Lingxian White (LX), Yangjiang (YJ), Yan (YE), Wuzong (WZ), Baizi (BZ), and Xupu (XP) geese (*Anser cygnoides*), have less than 1000 hens [6]. The efficiency and output values of genetic resources in some places are low, and high-yield breeds are often introduced for hybridization. Chaotic mating is a serious phenomenon that results in fewer pure breeds, and some existing goose breeds are close to extinction.

The genetic diversity of organisms is the basis for ensuring the survival and evolution of breeds and is of great significance for evolutionary polymorphism analyses, genetic relationship analyses, germplasm resource optimization, and the protection of existing breeds [7]. Currently, microsatellite markers and whole genome sequencing methods are used to study the genetic diversity and population structure of endangered goose breeds [8,9,10]. Wen et al. also used whole genome sequencing to reveal the origin breeds of Chinese domestic geese [11]. Most domestic goose breeds in China came from *Anser cygnoides*, whereas European domestic geese were derived from *Anser anser*. Only the Ili goose, which is distributed throughout Xinjiang, originated from *Anser anser* [11]. Heikkinen et al. selected 14 breeds (*Anser anser*) from the Eurasian continent and conducted the first genome-based inference using whole genome markers of European geese. They explain that the fixation index of European goose breeds (grayleg and domestic geese) is 0.15800 [12]. In these efforts, due to the limited number of known markers, the investigators used markers that originated from different species or breeds. This resulted in fewer alleles being labeled in many loci or the target sequence not being amplified via a polymerase chain reaction (*PCR*). Weiβ et al. reported that several microsatellite markers were isolated from grayleg geese (*Anser anser*) [13]. However, most of those markers revealed low levels of polymorphism in endemic endangered goose breeds. Therefore, the use of microsatellite markers and whole genome sequencing methods to investigate the genetic diversity of species has certain limitations. The mitochondrial genome has unique characteristics that distinguish it from the nuclear genome.

Mitochondrial DNA (mtDNA) has specific characteristics such as a simple structure, maternal inheritance, rapid evolutionary speed, and almost no recombination, making it the best molecular genetic marker for studying a species’ origin, evolution, and classification [14]. mtDNA is a type of DNA that exists in all animal cells. It is a circular genome with 16,569 base pairs (bp), which are passed down through maternal generations [15]. The mitochondrial genome has many copies of mtDNA, including 13 protein-coding genes, *22 tRNA* genes, *2 rRNA* genes, and non-coding regions. The mtDNA cytochrome b (*CYTB*) gene has a moderate evolutionary rate and is suitable for detecting genetic differences at the population level. It is an ideal marker for studying a population’s genetic structure and diversity [16,17]. Previous studies have mostly focused on the mitochondrial *D-loop*, *ND6*, and *COI* regions. Abdel-Kafy et al. used the *D-loop* region to study the phenotype and genetic characteristics of Egyptian geese and found that the potential heritability of the head, stem, tarsal length, and live weight was relatively low [18]. Jia et al. used the mitochondrial *ND6* region of chickens to explore the sequence combinations of several different regions between breeds, which can provide a more comprehensive and accurate understanding of the maternal origin of chickens [19]. Zhang et al. suggested that the mitochondrial *COI* region could be amplified to identify goose breeds [20]. There have been no systematic studies on the genetic diversity and evolutionary analysis of mitochondrial *CYTB* genes in the six locally endangered goose breeds (*Anser cygnoides*) included in our study.

In this study, blood samples from six locally endangered goose breeds were collected, and the DNA was purified. *PCR* amplification and Sanger sequencing were performed on *CYTB* mtDNA molecular markers to establish a DNA barcode database. The complete sequence of the mitochondrial *CYTB* gene has been used as a molecular marker to evaluate the genetic diversity of mtDNA and identify specific polymorphic sites [21]. The mitochondrial gene labeling method is expected to provide a theoretical basis for the identification, protection, breeding, and utilization of endemic endangered geese genetic resources in China.

## 2. Materials and Methods

### 2.1. Ethics Approval

All animal experiments were approved by the Institutional Animal Care and Use Committee of Yangzhou University (approval number: 132-2022; date: May 2022). All procedures were performed in accordance with the Regulations on the Management of Laboratory Animal Affairs (Yangzhou University, 2012) and the Standards for the Management of Experimental Practices (Yangzhou, China, 2008).

### 2.2. Experimental Design and Facilities

Based on the distribution of endangered breeds of geese in China, 180 specimens were collected from six geographical locations (Figure 1). The LX, YE, YJ, WZ, XP, and BZ breeds (*Anser cygnoides*) were selected for our study (Table 1 and Appendix A). After bathing the geese, an intravenous blood sample was collected from each of the 180 geese. All experimental geese were raised by the National Waterfowl Resource Conservation Bank (Taizhou, China). They were divided into six groups according to the breeds, with 30 birds in each group (*n* = 30 birds/group, 15♂ + 15♀). The blood samples were kept in test tubes containing an anticoagulant and stored at −20 °C until the DNA extraction.

Based on the complete sequence of goose (*Anser cygnoides*) mitochondria published on the NCBI website (GenBank: MN122908.1), a sequence comparison was conducted using DNASTAR Navigator 11 software and screened for conserved regions that could be used to amplify the entire *CYTB* gene to design appropriate primers (Figure 2). The specific primers designed to amplify the goose mitochondrial *CYTB* gene using Primer 5.0 are as follows: F: 5′-TAATCAACAACTCCCTAATCG-3′, R: 5′-TGAAGTTTTCTGGGTCTCC-3′ (Tm = 54.8 °C). Genomic DNA was extracted from the whole blood of each adult animal using a TIANamp Genomic DNA Kit (YDP304-03, TIANGEN Biotech, Beijing, China), according to the manufacturer’s protocol. PCR was performed using the Premix Taq (Takara, Beijing, China). For *CYTB* amplification, we used 1.5 μL of DNA (Concentration: 598.7 ± 24.25 ng/μL), 1.5 μL of upstream and downstream primers (10 pmol/μL), and 12.5 μL of 2× Taq PCR Mastermix, to which ultrapure water was added to make a final volume of 25 μL. Primer *CYTB-N*/*CYTB-J* sequencing was used for fragments of *CYTB* [22]. The polymerase chain reaction (PCR) protocol was as follows: 35 cycles at 94 °C with 3 min of initial denaturation, 30 s at 94 °C, 30 s at 54.8 °C, and 30 s at 72 °C, then extension at 72 °C for 5 min, following which the PCR instrument was turned off when the temperature dropped to 4 °C. PCR stock solution electrophoresis was performed in a 1.0% agarose gel to detect the non-specific presence of the band, and the target product was sent to Qingke Biotechnology Co., Ltd. (Nanjing, China) for synthesis and sequencing.

### 2.3. Sequence Analysis of Genetic Diversity

All raw sequencing data were processed using MEGA 11.0 software (Molecular Evolutionary Genetics Analysis 11.0) to achieve interindividual sequence alignment, correction, and shearing [25]. The mtDNA fragment sequence was visualized using Chromas2.4.1 software and manually browsed to ensure the accuracy of the bases. The haplotype (Hap), haplotype diversity (*Hd*), average number of nucleotide differences (*K*), and nucleotide diversity (*Pi*) were analyzed using DNAsp (DNA Sequence Polymorphism 6.12.0) for variable sites and compared using BioEdit. A pairwise identity matrix analysis of mitochondrial gene sequences was performed using SDT v1.2 software [26]. Analyses of population differentiation (*Fst*) and gene flow (*Nm*) were performed using Arlequin 3.5 with 1000 permutations. The *Fst* values range from zero to one, where zero indicates complete mixing [27].

### 2.4. Haplotype Network and Population Structure

A haplotype represents a group of alleles in an organism. The haplotype of mtDNA refers to a specific arrangement of alleles on mitochondrial DNA. Due to the fact that mtDNA is only transmitted in the maternal line, studying the haplotype of mtDNA can trace the genetic information of the maternal line. A haplogroup represents a group of similar haplotypes that share a common single nucleotide polymorphism ancestor [28]. Haplotype data for the combined sequences were edited using DNAsp 6.12.0. We calculated the distance between all breeds using MEGA v.11.0 (Kimura’s 2-parameter model and Tamura’s 3-parameter model) and then plotted the maximum likelihood (*ML*) evolutionary tree for the haplogroup of geese. Finally, the median-joining (*MJ*) network of the control region of mtDNA *CYTB* haplotypes was drawn using Popart v.1.7 software [27]. All data entries for haplotype samples are provided in the internationally recognized database NCBI (GenBank: PP515716-PP515892) and Appendix A. The specific distribution of purine and pyrimidine polymorphic sites was analyzed using Excel 2019 software for statistical haplotype sequence arrangement. The R-Studio 4.2.1 software (https://www.chiplot.online/, accessed on 10 January 2024) was used to construct a heat map of the phylogenetic evolutionary relationships [29]. To track the demographic changes in the main lineages of various groups, Arlequin 3.5 was used to perform neutrality and mismatch distribution analyses, and GraphPad Prism 8 was used to draw the basic mismatch difference analysis graph. In two neutrality tests, Fu’s Fs value and Tajima’s D value were used to evaluate the historical demographic expansion using 1000 simulated samples. Negative values for both tests indicate that the population has experienced expansion, whereas positive values indicate that the population has experienced a bottleneck [30].

## 3. Results

### 3.1. CYTB Gene Sequence Amplification and Verification

The genomic DNA was extracted from 180 blood samples of endemic endangered geese (*Anser cygnoides*), and the *CYTB* fragment of mtDNA was found to be positive when used as a template. The total length of the PCR amplification products detected via electrophoresis was 721 bp (Figure 3A). This was consistent with the expected target fragment size, and the band was bright and specific. The raw solutions of all PCR products were sent to Qingke Biotechnology Co., Ltd. (Beijing, China). for forward sequencing in the next step of the analysis.

### 3.2. Analysis of Genetic Structure of Endemic Endangered Goose Breeds

#### 3.2.1. *CYTB* Gene Locus Information and Nucleic Acid Diversity Analysis

DNAsp software (version 6.0) was used to perform a homology alignment of the gene sequences of the endemic endangered goose breeds, and misread nucleotides were trimmed, aligned, and corrected. The results of the mtDNA *CYTB* gene information site and nucleotide diversity analyses showed (Table 2 and Figure 3B) that the final length of the analyzed sequence was 707 bp. This study identified 117 polymorphic sites, accounting for approximately 16.55% of the total sequenced sites (117/707). Moreover, the A + T content of the sequence (51.03%) was slightly higher than the G + C content (48.97%), indicating a preference for purine bases (A/T). The YJ, YE, and WZ geese had the highest population genetic diversity, with more than 20 polymorphic sites detected (*NPIS* + *PIS*). The haplotype diversity was greater than 0.9 (*Hd* > 0.9), and the nucleic acid diversity was greater than 0.01 (*Pi* > 0.01). The average nucleotide difference between the YJ and WZ goose breeds was 8.01 (*K* > 8.01), which was much higher than the *K* values of the other four breeds.

#### 3.2.2. Comparison of Base Homology within CYTB Gene Sequence of Each Breed

A pairwise distribution analysis of the *CYTB* gene sequences within the endemic endangered goose breeds resulted in different thresholds for sequence classification, ranging from 97% to 98%, 99%, and 100% (Figure 4). At the threshold of population genotyping, each of the six endemic endangered goose breeds fell into one of three clades; among them, the YJ and WZ geese occupied a higher proportion of gene II, with a critical point of 97%, the BZ geese had the highest proportion of gene III within the breeds, and the three other goose breeds had basically the same proportion of genes I, II, and III, which are called evolved genes. Three evolutionary genes (I, II, and III) were clearly distinguished, with a demarcation threshold of 98%.

#### 3.2.3. MtDNA *CYTB* Gene Haplotype Distribution and Proportion Analysis

We used the symbiotic relationship between goose breeds to draw an intermediate connectivity network diagram for endemic endangered goose breeds (Figure 5A). We detected 81 haplotypes in the mitochondrial *CYTB* region based on nucleotide variations between sequences. The haplotypes of goose breeds exhibit a complex star-shaped divergent distribution. Using the maximum likelihood (*ML*) evolutionary tree to construct an evolutionary phylogenetic tree, all haplotypes were divided into six major branches, forming six genetic lineages (Figure 5B). Four haplotypes (Hap_4, Hap_10, Hap_13, and Hap_14) dominated the goose breeds, and fifty-one were singletons. The distribution and proportion of haplotypes unique to six endangered goose breeds are shown in Table 3. The haplotype distribution of the six breeds was biased towards females. In general, the LX and YE breeds exhibited a rich genetic diversity, each containing 23 or more haplotypes. The number of unique haplotypes within the LX and YE goose breeds was less than 10, accounting for 39.13% and 37.04% of the total haplotypes, respectively. The largest number of unique haplotypes was found in the WZ breed with 12, accounting for 60.0% of the total number of haplotypes. This was the only breed with a greater number of haplotypes within breeds than between breeds. XP geese had the second largest number of unique haplotypes, accounting for 47.83% of the total number of haplotypes. In addition, Hap_10 was shared by six goose breeds, accounting for 1.23% of all haplotypes.

#### 3.2.4. Analysis of Variable Loci of *CYTB* Gene in Endemic Endangered Geese

The distribution of the variable sites of various haplotypes among the breeds is shown in Figure 6 and Appendix A. A comparative study using mitochondrial *CYTB* sequences from *Anser cygnoides* (GenBank: GCF_002166845.1) revealed that all the haplotypes had a base deletion at position 54, whereas Hap_66 had a deletion at position 55 and position 54. Regarding the distribution of variable sites among breeds, after checking for base insertions and deletions, it was found that adenine (A) mutation sites accounted for the largest proportion of mutations at 43.33%. The proportions of mutations for cytosine (C), guanine (G), and thymine (T) were 30.00%, 26.67%, and 10.00%, respectively. Since all insertions and deletions were removed from the analysis, variable types were classified as transitions and transversions (G/A, T/C, A/G, and C/T), exhibiting a strong transversion bias in the mtDNA.

### 3.3. Analysis of Clustering Relationships of CYTB Genes in Endemic Endangered Geese

Based on the Kimura two-parameter model, we calculated the genetic distance among goose groups and utilized the unweighted pair-group method using arithmetic average (UPGMA) to generate a clustering heat map (Figure 7). The genetic distance between the BZ and YJ goose breeds was the greatest (*GD* = 0.267). Endemic endangered goose breeds can be roughly divided into two major clusters based on the genetic distances between breeds. Among them, the WZ and YJ geese were grouped together, and there was frequent gene flow between their ancestors. The BZ and XP geese were first grouped into one branch and then grouped together with LX and YE geese. The two large groups were clustered into one large group belonging to the same breeds. Based on BLAST analysis and GenBank data, the most likely species of *Anser cygnoides* were identified.

### 3.4. Genetic Differentiation Index and Gene Flow of Endemic Endangered Breeds of Geese

The Wright’s fixation index (*Fst*) and gene flow (*Nm*) were calculated for all six breeds (Table 4). The *Fst* value between each group was in the range of 0.004–0.991, which was consistent with the differentiation index range. Notable genetic differentiation was observed when comparing the LX, XP, and YJ breeds with the BZ breed, and a similar result was found when comparing LX with WZ geese (*Fst* > 0.25). Additionally, moderate genetic differentiation was observed when comparing LX and YE geese with the YJ breed, whereas the BZ and YE breeds were similar (0.05 < *Fst* < 0.15), and the degree of genetic differentiation between XP and YE, YJ, and WZ breeds was the smallest (*Fst* < 0.05). The *Nm* values showed a frequent genetic exchange of LX, YJ, and YE with WZ breeds (*Nm* > 15.00), indicating that inter-breed hybridization occurred owing to their geographical proximity. However, the gene flow index among the LX, XP, and BZ breeds was low (*Nm* < 1), indicating that gene flow among the three breeds was low because of geographical location factors. In particular, there was no genetic exchange between XP and YE geese (*Nm* = −62.45). The results obtained through the molecular analysis of variance (Table 5) showed that the genetic differences between breeds mainly originated from individuals (*p* = 0.001). For all breeds, the degrees of freedom, sum of squares, and percentage of variation among populations were significantly lower than those within breeds (*p* < 0.05). A total of 92.90% of genetic variations occurred within breeds. There has been no recent introduction or mating between breeds in China that has inhibited gene flow or led to the differentiation of haplotype branches in goose breeds.

### 3.5. Population Historical Dynamic Analysis

Neutrality tests and nucleotide mismatch difference analyses were performed based on the mitochondrial *CYTB* gene sequences. The Fu’s Fs values for the BZ, LX, XP, and YE geese neutrality tests were negative, and Tajima’s D value was positive. The values of these statistics were not significantly different from zero (*p* > 0.10) (Table 6). The Fu’s Fs value for the XP breed of geese was significantly different from zero (*p* < 0.02). Combined with the nucleotide mismatch distribution map (Figure 8), the expected values for the pairwise differences in genes in the control regions of the BZ, LX, XP, and YE geese presented a smooth curve, whereas the actual values all appeared to reach the highest peak. This showed that the distribution of nucleotide difference mismatches in the mitochondrial *CYTB* gene sequences of the four breeds presented a typical unimodal distribution, which was consistent with the unimodal curve pattern of population expansion and deviated from the neutral detection results. The Fu’s Fs values for the YJ and WZ breeds’ neutrality tests were close to zero, Tajima’s D value was positive, the difference was not significant (*p* > 0.05), and the two breeds were relatively stable. Combined with the nucleotide mismatch difference distribution map, while the expected values exhibited a smooth curve, the actual values for the two breeds displayed a multimodal Poisson distribution. The results of the Bayesian skyline plot (*BSP*) analysis showed that the group maintained an overall stable trend. It was speculated that the WZ and YJ breeds were relatively stable as a whole and had not experienced population expansions or bottleneck effects, which was consistent with the neutrality test results.

## 4. Discussion

Biodiversity includes genetic, reproductive, and ecological diversity [31]. Research on the global genetic diversity of organisms is a prerequisite for exploring the evolution of life and reproductive diversity. The genetic diversities of various animals can be analyzed at the genetic and molecular levels [32]. By further studying genetic diversity using molecular markers, genomic DNA markers (microsatellites, internal transcription spacer 2 [*ITS2*], and mitochondrial markers [*cytochrome b*, *COI*, and *ND4*]), we can reveal the variations or genes involved in the phenotypic changes of different breeds that may rapidly evolve after domestication, forming specific phenotypic characteristics. Regions or loci that have undergone selection exhibit specific characteristics, including high population differentiation, significantly reduced nucleotide diversity levels, and long-range haplotype homozygosity [33,34]. Therefore, based on these principles, we studied different parameters by measuring specific sites in the mtDNA *CYTB* region. The highest haplotype diversity and average nucleic acid difference detected within the six breeds of geese included in our study were 0.993 ± 0.011 and 8.463 ± 0.202, respectively. Li et al. determined the mtDNA *D-loop* sequences of 26 goose breeds *(Anser cygnoides*) and six *Lande* geese (*Anser anser*), with an average *Hd* of 0.1384 and *Pi* of 0.00029 for common breeds of Chinese geese [35]. The genetic diversity of endemic endangered goose breeds is considerably higher than that of other goose breeds (*Anser cygnoides*). Therefore, we believe that the mitochondrial *CYTB* region has accurate maternal inheritance, a conservative genetic structure, a moderate evolutionary rate, and is an effective molecular marker for studying the genetic structure of endangered waterfowl. This may be owing to the differences in the specific sites of each gene-coding and non-coding region in the mtDNA fragment. When the obtained *CYTB* sequence of six goose breeds was compared with data from GenBank, *Anser cygnoides* was identified using BLAST. A study of greylag geese (*Anser anser*) using 204 base pair fragments of mitochondrial control regions showed that the purine (A + T) content was slightly higher than that of pyrimidine (G + C) [36]. This phenomenon was consistent with our results. It is possible that the nucleotide encoding the protein causing the mutation was subject to less natural selection pressure at the codon site. In the intraspecific sequence comparison of endemic endangered goose breeds, the similarity between partial sequences of individuals was maintained at thresholds of approximately 97%, 98%, and 100%. This result was similar to the experimental results obtained by Ran et al. for mountain plum chickens [37]. In our research, we chose a specific gene segment (the *CYTB* region) rather than the entire mitochondrial sequence for sequencing analysis. Therefore, measuring longer sequences between different genes within the entire mtDNA sequence may provide more comprehensive information on specific mutation sites than measuring shorter sequences, a hypothesis that requires further testing. Based on the genetic diversity information of endemic endangered geese excavated, we selected geese with ideal mutation site sequences as breeding targets and formulated corresponding protection and restoration plans, including the establishment of artificial incubation centers and implementation of habitat restoration plans, to increase the population size and maintain genetic diversity. These results provide further data for optimizing the population structure of endemic endangered breeds and improving breeding methods.

The assembly of haplotype genomes is of great significance for the analysis of structural variation between haplotypes, the study of the evolution of breeds’ genetic origins, the evolution of sex chromosomes, deleterious mutations, and the exploration of the molecular mechanisms of hybrid vigor [38,39]. The number of mtDNA haplotypes at trapping sites positively correlates with the number of individuals examined per population [40]. Although mtDNA is inherited maternally, the number of haplotypes in males is not closely related to the number in females [41]. In this study, we collected blood samples from 180 geese, and 81 haplotypes were detected based on nucleotide variation between sequences (a high proportion of female haplotypes). A phylogenetic tree constructed from *CYTB* gene haplotypes showed that the breeds could be divided into six major branches, indicating limitations in their genetic lineage. A study of *Neocaridina denticulata* showed that the phylogenetic relationships between different groups are cross-nested and did not show distinct geographical aggregation [42]. This indicates that haplotype differentiation between the groups was not apparent. Owing to the significant differentiation of haplotypes within poultry breeds, the results of our study contradict those found in poultry breeds [37]. We found that the number of unique haplotypes in YE, XP, and WZ geese exceeded 10 (*N* > 10). Haplotype sharing was observed between samples collected from different breeds, except for Hap_10 (1.23%). Other endemic endangered breeds formed relatively distinct geographical structures, and groups in different geographical locations mostly had unique haplotypes. The results for the variable sites of the different haplotypes showed that Hap_4 and Hap_10 were the dominant haplotypes among the groups (*N* > 12). This may be owing to the widespread genetic mixing between and within goose breeds; moreover, human factors cannot be ruled out (artificial introduction of foreign genes and isolation and protection). After checking for base insertions and deletions, the adenine (A) mutation site was found to be relatively high (43.33%), indicating a strong transversion bias characteristic of animal mtDNA. The conclusion reached by Boman et al. [43] is contrary to that of our study. The abnormal situation may be owing to short evolutionary time constraints; both female relatedness and limited male-mediated gene flow significantly reduce mtDNA genetic variation in breeding colonies or small breeds [44]. Second, insertions and deletions during sequence processing alter the sequence of mitochondrial DNA, resulting in mutations that require further study on the variable types of transitions and reversals. In summary, our research has found that large geographical distances and human activities jointly restricted the diffusion of haplotypes in the populations of geese under study, thereby increasing the differentiation of the north–south haplotype branches of six endemic endangered breeds.

The degree of difference in the average genetic distance and between breeds reflects the distance of genetic relationships between various groups or the genetic similarity between breeds [45]. In our research, BLAST sequence comparison and inter-variety clustering analysis of 180 samples showed that the six goose breeds were roughly divided into two large evolutionary groups. BZ, XP, LX, and YE geese were divided into three subgroups. These were all local breeds in China; therefore, they were grouped into one major category. BLAST analysis generated all 81 different haplotypes, and the most likely species of *Anser cygnoides* were identified based on GenBank data. Our results were consistent with those of previous studies that used mitochondrial data to determine the origins of domestic geese in China and Europe [11,46]. The larger the value of *Fst*, the greater the genetic diversity of the breeds and the higher the degree of population differentiation [47]. We observed the relationships between locally endangered goose breeds. The LX, XP, and BZ geese had the highest degree of genetic differentiation (*Fst* > 0.8), and the gene flow values were not significant (*Nm* < 1). There is substantial genetic differentiation between breeds when *Fst* ≥ 0.33 [48]. Generally, the genetic exchange between the LX, XP, and BZ geese was low compared to that of the other three breeds. Substantial genetic differentiation caused by genetic drift may have occurred, which is consistent with the results of the genetic distance analysis. The degree of genetic differences between the LX, YJ, YE, and WZ goose breeds was relatively low (*Nm* > 15.00). Frequent gene exchange can effectively inhibit genetic drift and reduce the risk of genetic differentiation among groups [49]. However, the overall genetic exchange between groups was not smooth and was significantly affected by geographical isolation. No significant difference was observed between breeds that maintained a high level of genetic diversity (*p* > 0.05). In this study, the Wright’s fixation index between endemic endangered goose breeds in China was close to zero (*Fst* = 0.07099), and the genetic structures of the different breeds were completely consistent. Therefore, we speculate that this may be owing to the gradual expansion of the scale of artificially raised geese in northern China, using excellent male breeds (*Lande* and *Rhine*) for hybrid breeding [50]. Frequent human activities promote male-mediated gene flow in adjacent breeds (such as setting up corridors), thereby promoting the dispersal and exchange of goose breeds.

The fact that the variability in a breed’s genetic diversity is smaller than the census population size among breeds is known as Lewontin’s paradox [51]. The reason for this scaling is unclear; however, it is likely that multiple factors are involved (over collection, habitat fragmentation, and accidental events). To determine whether a physiological population has experienced population expansion, we used two dynamic detection methods: nucleotide mismatch difference analysis and neutrality tests. Expansion events result in smaller genetic differences between most individuals, because they are mainly derived from a small group of ancestral breeds. In this case, the distribution of nucleotide mismatch differences exhibits a single peak “Poisson distribution” characteristic. Tajima’s D test is more likely to reveal the history of ancient population expansion, whereas Fu’s Fs test is more sensitive to recent population expansion [52]. In our study, the Fu’s Fs of the neutrality test for the BZ, LX, and YE geese was negative, and Tajima’s D was positive. However, there was a lack of statistical significance in the theoretical sense, thus deviating from the neutrality test results. Similar results were obtained using a diversity analysis of slender fish breeds in the Yalu River Basin [53]. Combined with the nucleotide mismatch distribution map of the six goose breeds, the expected values for pairwise differences in the *CYTB* control region showed a smooth curve, whereas the actual values had the highest peak. Nucleotide mismatches in the mitochondrial *CYTB* sequences of the BZ, LX, YE, and XP breeds exhibited a typical unimodal distribution. They conformed to the unimodal curve pattern of population expansion, and it was inferred that the BZ, LX, YE, and XP breeds had recently experienced population size expansions. This pattern may be caused by human intervention in the introduction and breeding of endemic endangered breeds and the demarcation of nature reserves for endemic endangered breeds to protect their natural reproduction from environmental threats. The increased occurrence of extreme weather events caused by global warming and climate change might affect the population fluctuation patterns of endemic endangered breeds [54]. This study found that the neutrality test Fu’s Fs of WZ and YJ geese were close to zero, and Tajima’s D was positive. The two groups were relatively stable and consistent with the neutrality test results. The nucleotide mismatch difference distribution map showed that the expected values for pairwise differences in genes in the population control region showed a smooth curve, whereas the actual values in the two breeds showed multimodal Poisson distributions. We speculate that the breeds WZ and YJ were relatively stable and did not experience population expansions or bottleneck effects. These finding differs from those of previous studies [55]. Sampling points reflect the geography, potentially influencing the temperature, environment, and local adaptability. Furthermore, differences in the climate and environment cause changes in the local adaptability. Human factors such as culling and release have caused differences between endemic endangered groups in different geographical locations and habitats [56]. The current climate may cause changes in volatility cycles, affecting the dispersal patterns of WZ and YJ geese. Future research should aim to compare the genetic diversity observed through hybridization and different molecular markers (including genomics) between current and future breeds at the same research sites.

## 5. Conclusions

In summary, our research indicates that compared to other mitochondrial regions (*COI*, *ND6*, etc.), the *CYTB* control region was more helpful in further understanding the genetic diversity and population structure of goose breeds. A total of 81 haplotypes with multiple genetic lineages were detected in six locally endangered goose breeds in China. Through BLAST analysis and obtaining different haplotypes, based on GenBank data, these six breeds were identified as the most likely breeds of *Anser cygnoides*. However, there was no significant differentiation between the various breeds, maintaining a low level of genetic diversity. In addition, the Wright’s fixation index of endemic endangered goose breeds in China is significantly lower than European goose breeds. Therefore, six endangered goose varieties are facing significant extinction risks. Due to the Chinese government’s insufficient strategies and technologies for protecting animal species, it remains imperative to persist in safeguarding the genetic diversity of breeds of geese and establish a comprehensive framework for further action. Our data indicate that endemic endangered geese, which are on the brink of extinction in some regions of China, represent crucial and diverse hot spots. Therefore, there is an urgent need to plan and protect the genetic resources of Chinese breeds of geese as well as develop appropriate strategies to maintain genetic variation.

## Figures and Tables

**Figure 1 animals-14-01480-f001:**
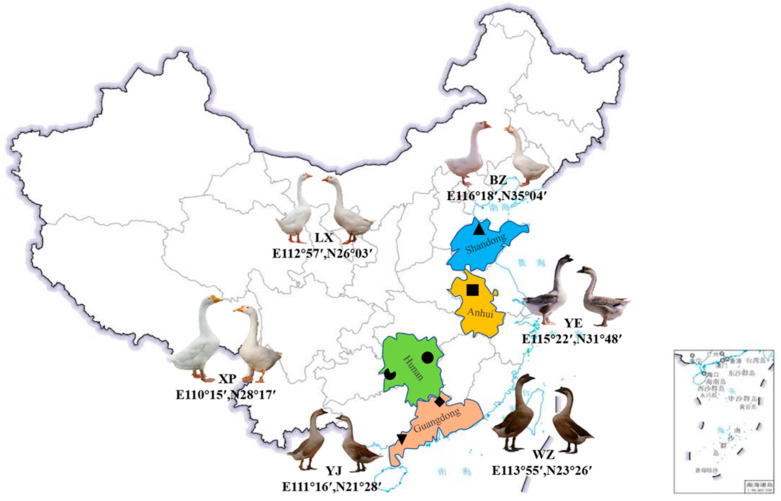
The six geographic collection points for the endemic endangered geese. Images of different goose breeds were captured using a digital camera (Osaka, Japan). LX, Lingxiang White; YE, Yan; YJ, Yangjiang; WZ, Wuzong; XP, Xupu; BZ, Baizi. The irregular black shapes (rectangle, triangle, etc.) in the figure represent the collection locations of the experimental animals, the rectangle in the bottom right corner is a part of Chinese territory.

**Figure 2 animals-14-01480-f002:**
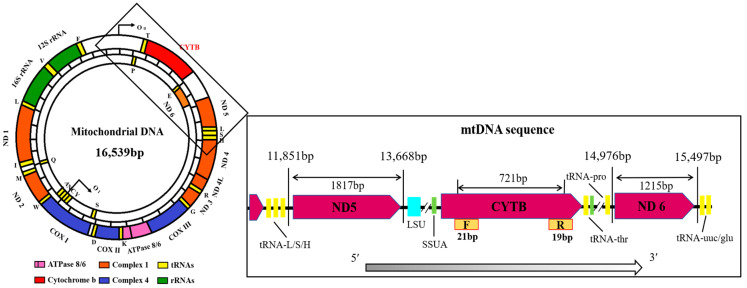
Circular structure and linear organization of goose mtDNA [23,24]. The sequence and transcription direction of *CYTB* and the adjacent coding regions are indicated by arrows. Fragments of SSU (green) and LSU (light blue) rRNA were detected between the protein-coding regions and aligned identically to fragments of chicken and duck breeds.

**Figure 3 animals-14-01480-f003:**
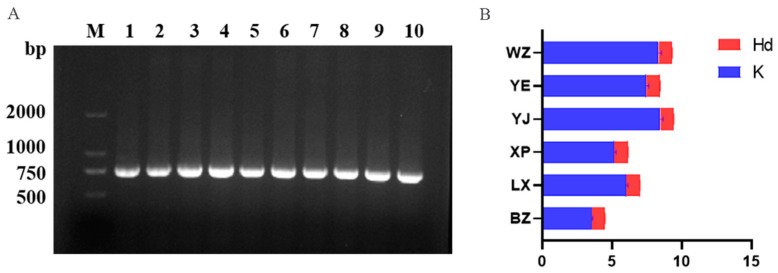
(**A**) Electropherogram of *PCR* amplification of the *CYTB* fragment. M: denotes the relative molecular mass standard (DL2000 marker); 1–10 indicate the positive products of goose *CYTB* gene amplification. (**B**) Nucleic acid diversity ratio of endemic endangered goose breeds. LX, Lingxiang White; YE, Yan; YJ, Yangjiang; WZ, Wuzong; XP, Xupu; BZ, Baizi; Hd, haplotype diversity; K, average number of nucleotide differences.

**Figure 4 animals-14-01480-f004:**
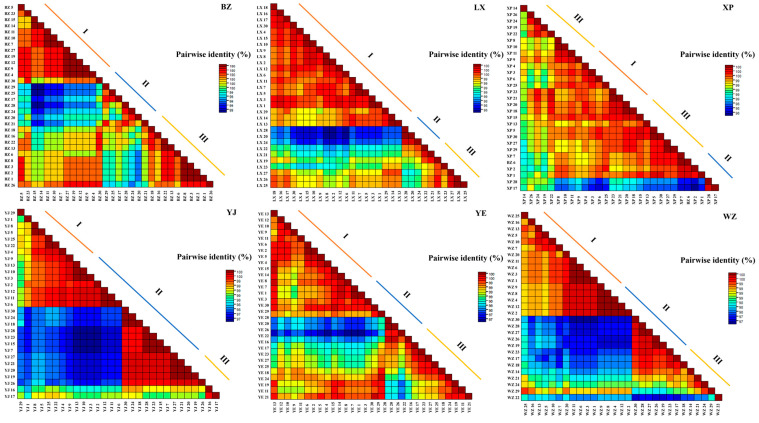
Pairwise identity matrix of mtDNA *CYTB* gene. Note: I, gene I; II, gene II; III, gene III.

**Figure 5 animals-14-01480-f005:**
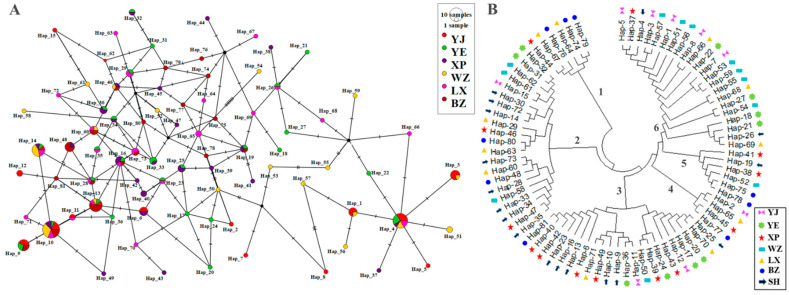
(**A**) Median-joining network diagram constructed based on *CYTB* gene haplotypes. Each circle represents a unique haplotype, the color represents endangered geese of different breeds, and the size of the circle is proportional to the number of isolates contained. The lines (shaded markers) on the branches indicate the location of the mutation, with one line for each mutation. (**B**) A phylogenetic tree constructed based on haplotype nucleotide sequence types. The symbols of different colors represent haplotypes only found in their respective breeds. SH represents the common haplotype of endemic endangered goose breeds, and the specific distribution is shown in Appendix A; all haplotypes are divided into six major branches, forming six genetic lineages.

**Figure 6 animals-14-01480-f006:**
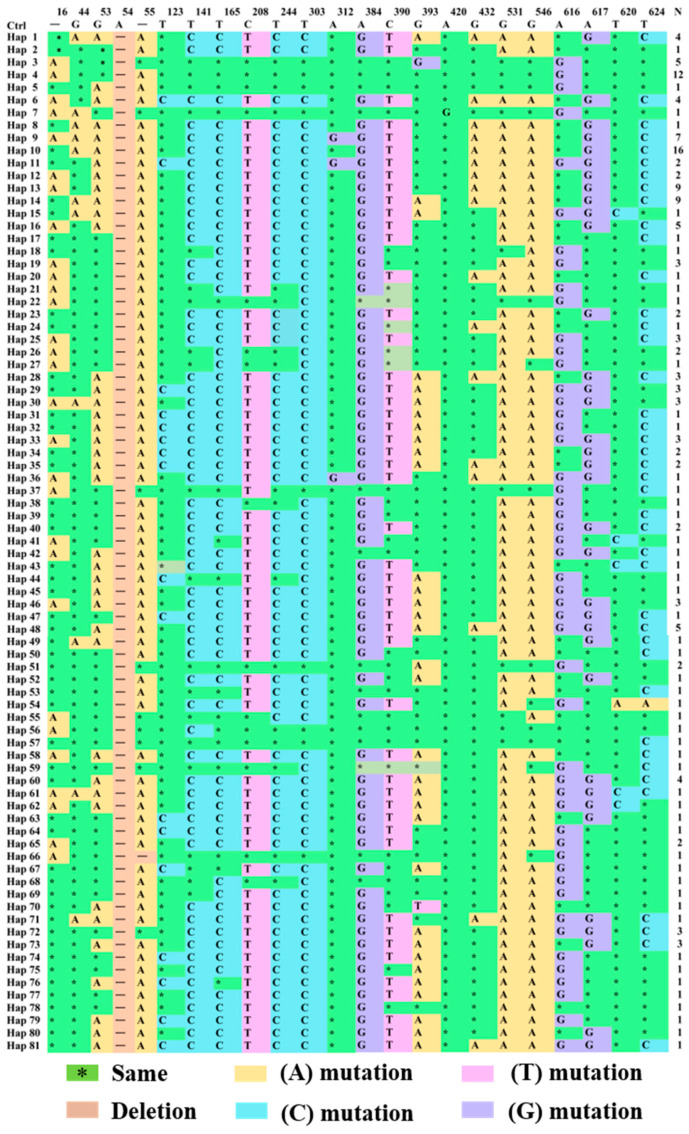
Variable site distribution controlled by mtDNA *CYTB* gene haplotype.

**Figure 7 animals-14-01480-f007:**
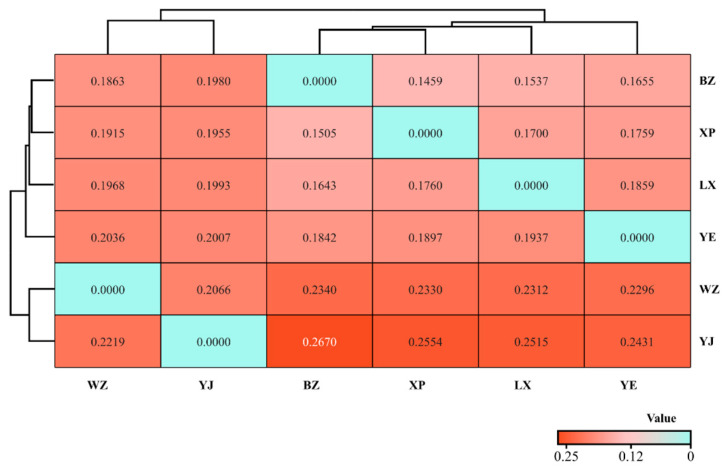
Evolutionary heat map of genetic distance relationship of mtDNA *CYTB* gene.

**Figure 8 animals-14-01480-f008:**
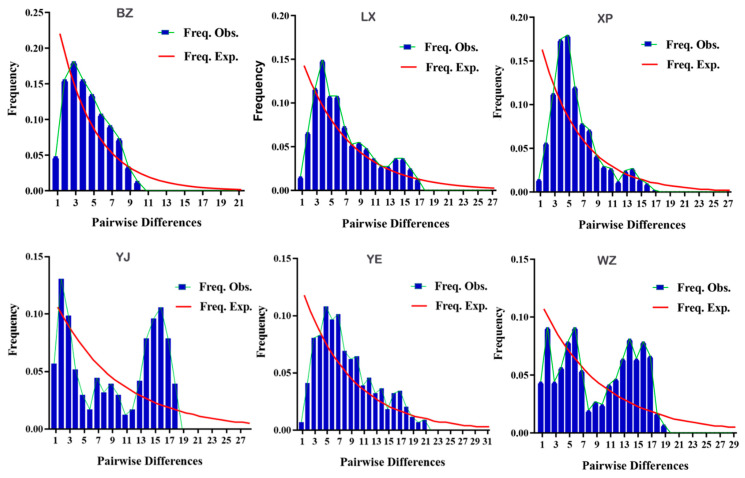
Base difference mismatch distribution map of mtDNA *CYTB* gene sequence. Note: Freq. Obs. represents the actual value; Freq. Exp. represents the expected value.

**Table 1 animals-14-01480-t001:** Local goose breed resource information table.

Breed	Birthplace	Appearance Characteristics	Number/w	Protection Level
LX	Zhuzhou, Hunan	Compact body shape and pure white feathers	0.04	Endemic endangered
YE	Lu’an, Anhui	Flat sarcoma with black and gray feathers	0.09	Endemic endangered
YJ	Yangjiang, Guangdong	Large body size and pure white feathers	0.09	Endemic endangered
WZ	Qingyuan, Guangdong	Black mane with back feathers and black-gray feathers	0.20	Endemic endangered
XP	Xupu, Hunan	Head sarcoma and black-gray feathers	0.60	Dangerous
BZ	Jinxiang, Shandong	Wide and short body with pure white feathers	1.00	Dangerous

Note: LX, Lingxiang White; YE, Yan; YJ, Yangjiang; WZ, Wuzong; XP, Xupu; BZ, Baizi. W represents 10,000 geese; for instance, the number/w = 0.04 indicates that there are currently 400 LX geese in China.

**Table 2 animals-14-01480-t002:** *CYTB* gene locus information, haplotype, and nucleotide diversity.

Breed	NPIS	PIS	MS	GC	Hd	Pi	K
BZ	1	10	646	48.87%	0.952 ± 0.0230	0.00538 ± 0.00054	3.538 ± 0.081
LX	1	17	642	48.97%	0.984 ± 0.0120	0.00910 ± 0.00162	6.005 ± 0.138
XP	0	18	621	48.94%	0.985 ± 0.0120	0.00806 ± 0.00106	5.153 ± 0.123
YJ	4	18	629	48.99%	0.943 ± 0.0200	0.01300 ± 0.00060	8.463 ± 0.202
YE	3	20	633	49.27%	0.993 ± 0.0110	0.01137 ± 0.00112	7.457 ± 0.171
WZ	8	17	614	48.78%	0.956 ± 0.0240	0.01307 ± 0.00199	8.350 ± 0.199

Note: NPIS: non-parsimony informative loci; PIS: parsimony informative sites; MS: monomorphic sites; GC: GC content; Hd: haplotype diversity; Pi: nucleotide diversity; K: average number of nucleotide differences.

**Table 3 animals-14-01480-t003:** Number and distribution proportion of haplotypes of endemic endangered goose breeds.

Breed	h	Unique Haplotypes	Proportion
BZ	19	7 ♀: Hap_74, Hap_75, Hap_76, Hap_77, Hap_78, Hap_79, Hap_80	42.11%
		1 ♂: Hap_81	
LX	23	7 ♀: Hap_63, Hap_64, Hap_65, Hap_66, Hap_67, Hap_68, Hap_69	39.13%
		2 ♂: Hap_70, Hap_71	
XP	23	6 ♀: Hap_37, Hap_38, Hap_39, Hap_40, Hap_41, Hap_42,	47.83%
		5 ♂: Hap_43, Hap_44, Hap_45, Hap_47, Hap_49	
YJ	15	4 ♀: Hap_2, Hap_5, Hap_7, Hap_8	40.00%
		2 ♂: Hap_12, Hap_15	
YE	27	7 ♀: Hap_17, Hap_18, Hap_20, Hap_21, Hap_22, Hap_24, Hap_27	37.04%
		3 ♂: Hap_31, Hap_32, Hap_36	
WZ	20	8 ♀: Hap_50, Hap_51, Hap_52, Hap_53, Hap_54, Hap_55, Hap_56, Hap_57,	60.00%
		4 ♂: Hap_58, Hap_59, Hap_61, Hap_62	

Note: h, number of haplotypes; LX, Lingxiang White; YE, Yan; YJ, Yangjiang; WZ, Wuzong; XP, Xupu; BZ, Baizi; ♀, female; ♂, male.

**Table 4 animals-14-01480-t004:** Genetic differentiation index *Fst* (below the diagonal) and gene flow *Nm* (above the diagonal) between breeds.

Breed	BZ	LX	XP	YJ	YE	WZ
BZ		0.090	0.000	1.400	6.220	1.390
LX	0.852		−27.140	3.840	−22.730	15.860
XP	0.991	0.019		2.770	−62.450	7.330
YJ	0.263	0.115	0.153		3.050	70.330
YE	0.074	0.023	0.008	0.076		31.880
WZ	0.153	0.031	0.064	0.004	0.015	

Note: BZ: Baizi; LX: Lingxiang White; XP: Xupu; YJ: Yangjiang; YE: Yan; WZ: Wuzong; *Fst*: Wright’s fixation index; *Nm*: gene flow.

**Table 5 animals-14-01480-t005:** Analysis of the genetic variation in the mtDNA *CYTB* sequence.

Breed	DF	SS	MS	EVV	PV	*Fst*	*p* Value
AP	5.0	51.483	67.354	0.24179	0.071	0.07099	0.001
WP	171.0	541.099	32.646	3.16432	0.929	-	-
Total	176.0	592.582	100.000	3.40611	1.000	0.07099	0.001

Note: AP: among breeds; WP: within breeds; DF: degrees of freedom; SS: sum of squares; MS: mean squares; EVV: estimated variation value; PV: percentage of variation; *Fst*: Wright’s fixation index.

**Table 6 animals-14-01480-t006:** Analysis of historical dynamics of endemic endangered goose breeds.

Item	BZ	LX	XP	YJ	YE	WZ
Fu’s Fs	−11.559	−13.025	−15.668	−1.020	−19.317	−5.615
*p*-value	*p* > 0.10	*p* > 0.10	*p* < 0.02	*p* > 0.10	*p* > 0.10	*p* > 0.10
Tajima’ D	0.88243	1.10756	0.43066	1.80847	1.00525	0.93890
*p*-value	*p* > 0.10	*p* > 0.10	*p* > 0.10	0.1 > *p* > 0.05	*p* > 0.10	*p* > 0.10

## Data Availability

All data generated or analyzed during this study are included in this published paper.

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
