# Peer review of "Analysis of Genetic Diversity and Population Structure of Endemic Endangered Goose (Anser cygnoides) Breeds Based on Mitochondrial CYTB"

_animals, 2024, doi:10.3390/ani14101480_

Round 1

Reviewer 1 Report

Comments and Suggestions for Authors

General comments:

Genetic diversity of organisms is the basis for ensuring species survival and evolution, and is of great significance to germplasm resources conservation and exploitation. In this study, the mitochondrial Cytb gene fragments of six endangered and Dangerous goose breeds were sequenced, and their genetic structures were compared and evaluated. This study maybe contribute to the understanding of the diverse genetic origins and historical dynamics of 6 goose breeds, and thereby preserve and exploit them in the future. Some suggestions are as follows:

1. In the abstract, briefly list the full names and abbreviations of the six endangered goose species.

2. In lines 278 and 279, "The genetic differentiation index (Fst) and gene flow (Nm) were calculated for all six populations (Table 4 and Table 5) " => "The genetic differentiation index (Fst) and gene flow (Nm) were calculated for all six populations (Table 4)". In addition, the results in Table 5 were not explained anywhere, please describe it, or remove it from the manuscript.

3. In lines 213 and 214, "XP geese have the highest proportion of genotype III within the species", however, as observed in Figure 4, BZ appears to have the highest proportion of genotype III within the species. Please check it.

4. In lines 313 and 315, "The Fu’s Fs values of the YJ and WZ…, Tajima’s D value was positive, the difference was not significant (P >0.10), …", but in Table 6, Tajima’s D value of YJ is 0.1>P>0.05, this is contradictory, please check the consistency of the listed data with the result description again.

5. Most of Figures in the manuscript are not so clear. Please replace them with high-resolution Figures.

Reviewer 2 Report

Comments and Suggestions for Authors

general comment

Please, choose one form of names, pops/breed and punctuation marks. It's really disturbs in reading.

It will disturb readers of related fields or guys who are unfamiliar with the topic.

Abstract is formatting pretty well, but the more you get into it formatting is annoying:

·         you should use space before and after: <, >, = (not “Nm=-62.45” but “Nm = -62.45”; not “Fst <0.3: but “Fst < 0.3”)

·         if you write “K > 8.01”, so keep going, but sometimes you use normal text sometimes italics for symbols in the text

·         till line 38 is fine with “ ‘ “ symbol, but after that you use “Tajima`s D” - that's not correct an apostrophe

·         as well in table 3 you are not used comma “ , “ but “ ” – it is not a comma

·         write gene names in italics! You are write for example: “Cytb”, “Cytb”, “CYTB”. one form, italic.

Please, use only one form – sometimes you write “CYTB Gene”, sometimes “Cytb gene”. In GenBank: MN122908.1 is “CYTB Gene”. Please choose one form. And italic.

Should be “Breed” instead “Item” in all tables. Please choose one form. In table 1 is “Breed”, in table 2 is “Item”, table 5 is “Source” and inside the table is “Pops”.

Provide data entries in internationally recognized databases (with accession numbers) of all haplotypes. Add these data to the supplement, add references in the text.

Abstract

line 36

“LX” – The full name of the breed does not appear before – provide the full name of the breed.

Introduction

line 50

“at the source of my country's livestock industry” – there is more than one author, should be “our”.

line 72-73

as Bos mutus 72 [7], insects [8], sheep [9], and Gekko japonicus

should be

as Bos mutus 72 [7], insects [8], sheep [9], and Gekko japonicus

line 105

“my country” – there is more than one author, should be “our”.

Materials and Methods

line 128

What is “AL/W” in Table 1?

Add description.

line 146

Please provide source graphics.

line 173

not “Mega” but “MEGA” (should be the same in whole text)

line 174-175

Please, specify the R package used in R software.

Results

line 205-207

·         “k: Average number of nucleotide differ” – should be big K consistently – “K”.

·         Note: The headline abbreviations explained should be in the order in which they appear in the headline.

·         Value for YJ – Pi is “0.01300 ± 0.0006” and should be “0.01300 ± 0.00060”

the same accuracy throughout the table - I don't know if something got cut off by accident, or if there should be zero there.

·         legend - either alphabetically according to abbreviations or according to the order in the table header; you have to look hard

line 241

Lack of the legend for Figure 5B.

line 242-246

“Note: Each circle represents a unique haplotype, the colour represents the country of origin of the genome, and the size of the circle is proportional to the number of isolates contained. The lines (shaded markers) on the branches indicate the location of the mutation, with one line for each mutation.”

·         This is description only for Figure 5A – it should be marked or the text should be before “(B)A phy-……” in description of Figure,

·         Check marks in Figure 5B – number of marks is not corresponding with name of Haps. Really Hap-35 is belong to second branch if you marked with red star instead yellow triangle?

·         These are just haplotype affiliations. There is no breed designation on Fig. 5B.

·         Add legend for Fig. 5B.

line 251

Is “(Hap_4 and 10)” and Should be “(Hap_4 and Hap_10)”

line 251

not “707bp”, but “707 bp”

for example 269 line and 290 line

“GD=0.267” – should be spaces before and after “=” – please review in all document.

The same in this para with lack of space after “>” – please review in all document.

line 276

“MtDNA” should be with small letter: “mtDNA”.

line 295

Should be the same accuracy throughout the table. Not “0.09” and “0.00” for example,  but 0.090 and 0.000, and so on in whole table – accuracy to third place in the entire table!

line 298-299

·         The same. Accuracy should be the same in the same column.

·         What are they: “Va” and “Vb” in EVV values? Add description.

Discussion

line 338

not “Li” but “Li et al.”

line 338

“D-Loop”. Should be “D-loop”

line 366-367

not “men and women” but “males and females”

line 369

not “Ma” but “Ma et al.”

line 238, line 374 and 378

You write “Hap10” and after that “Hap_4 and Hap_10”

But in table 3 you use without “_”.

Use one form in whole text and supplement!

line 394

“In this study, the results of the inter-species cluster analysis based on statistical genetic distance showed that the 6 endangered goose groups were roughly clustered into two evolutionary clusters. Geese from WZ and YJ were grouped separately because of their geographical proximity.”

Lack of evidence!!!

correct Fig. 5, add legend, assignment of haplotypes to breeds

especially fig 5 b - grouped by haplotype colours, not breeds - it is not known how breeds are grouped

line 348

not “Souza” but “Souza et al.”

line 399

“Anser cygnoides” – italics!!

line 390-424

Please discuss Fst with other breeds of endangered geese from the Anatidae family, from other countries.

Conclusions

line 467-481

Add information: are Chinese geese at greater risk of extinction than endangered geese in other countries?

Reviewer 3 Report

Comments and Suggestions for Authors

Dear Authors,

I read your manuscript about the research of goose breeds in China using mtDNA Cytb. In general, I was convinced that your work have enough importance to be published in MDPI Animals, however, only after major revisions of the manuscript. As you will see from various comments and suggestions below, major revisions are truly necessary in almost whole current manuscript. Not all revisions are of equal importance. Let’s start with the major problems that must be fixed. Firstly, it is not enough information presented regarding your research object. Is it 6 breeds or 6 different species? Why in the whole manuscript these goose species are not indicated? You need to present this information to readers. In case particular species of each of 6 breeds are unknown, clearly state it and give names of expected/presumed species. Secondly, it is really necessary to rewrite most introduction and part of discussion so readers can clearly see from the introduction what scientific progress currently exist regarding Anser genus, especially your studied breeds, genetic research. Consequently, you should mention all DNA molecular markers, as well present information whether genomes (not only mtDNA) of Anser genus species already known, and then must defend your choice to use mtDNA Cytb for your study in this context. Despite it is only single molecular marker (even whole mtDNA could be considered as one locus) reflecting maternal line in the age of genomics, I see no problem to publish your data if you will be able to defend your position regarding Cytb use of this research of 6 goose breeds/species genetics. Thirdly, your obtained mtDNA Cytb sequences reflecting distinct haplotypes should be deposited in GenBank and accession numbers should be provided in the updated manuscript version. Fourthly, if you want to use not only haplotype but also haplogroup conception then you must define particular criteria (preferably in methods) for haplogroups definition and this should be done using not only NJ (?) tree but also MJ haplotype network. Fixing of these four major problems is minimum before I can suggest to accept your work to be published in MDPI Animals. Once again, I tried to be objective and help you to upgrade manuscript to necessary level to be published so be encouraged and carry out necessary revisions. Below I presenting various comments and suggestions that I hope will be useful to you.

Simple Summary

In general, this part should be understandable to non-specialists. Currently it would be quite confusing for readers to read.

Line 16: ‘Population structure’ - give to readers more information regarding it: age structure, morphometric structure, phenetic structure, genetic structure and etc.

Line 18: ‘Fragments’ change to ‘fragment’, as only Cytb used.
Lines 19-21: Unsupported claim/statement, as number of haplotypes do not suggest migration among goose breeds. Haplotypes that were detected among all breeds can suggest that.

Lines 21-22: This is simple summary and not discussion, so try to elaborate everything that each reader clearly understand the value of your results, and how they can help to identify trade patterns.

Abstract

In general, this text can be left as it is with just small modifications but it still would be worth to upgrade it based on whole manuscript changes after revision process.

Lines 26-27: Remove part ‘at least…’ from sentence, as the current manuscript do not provide enough information about previous goose breeds studies using various molecular markers, and why mtDNA studies should be expanded. Alternatively, leave this sentence as it is but after major revision make sure that enough information was provided in the introduction regarding mentioned issues about molecular markers in goose breeds genetic research.

Line 35: Haplotype network is complicated, thus ‘central star-like distribution’ statement is not correct.

Line 36: ‘Among them’ - haplotypes, haplogroups, samples, else? Avoid confusion.

Introduction

In general, first paragraph requires some modifications while second and third paragraphs must be rewritten based on noted second major problem in the manuscript.

Line 48: ‘Chip’ needs more precise explanation to readers.

Lines 48-50: ‘my country’s’ change to China or broader meaning, as this is not local but international audience.

Lines 50-52: Selection is one of the processes/parts of microevolution/evolution, thus this word is not necessary or indicate it as not in the same level as evolution conception.

Lines 52-54: It would be correct to provide references for these claims.

Lines 54-57: What about two not mentioned species (28/30)?

Lines 68-69: mtDNA explanation is not good. Need to mention mitochondria, maternal line, whole size of Anser mtDNA (>16 kb). Lacking references.

Materials and Methods

In general, small but important changes are necessary. In 2.3. part should be mentioned K as well. Trimming of sequences also should be mentioned in the methods and not only in results.

Line 117: Provide goose species in latin in text or even better in Table 1. Also explain what is AL/W in Table 1.

Line 119: ‘subsequently’ - maybe you wanted to write ‘consequently’ or ‘actually’ or other word that introduce readers to more detailed description of the sampling?

Line 129: What goose species? Write latin species name also.

Line 135: Muscle tissue? Previously you mentioned only collection of blood from 180 specimen. Avoid such confusion to readers, as it create additional questions.

Line 138: DNA concentrations should be presented also.

Line 141: Try to avoid to use : two times in the same sentence/line.

Line 150: What kind of homologous species?

Results

Change everything based on noted fourth major problem and comments provided here, including lines, and results part of the updated manuscript should be good then. Do not know whether with your used software to create haplotype network is possible to use additional maximum parsimony option. This is possible with Network program. Then it is much easier to define haplogroups, as only most important lines among haplotypes are left. By the way, it might be worth to add additional sequences from GenBank to newly constructed haplotype network. Try to find more obvious image instead of Figure 3A, as at the moment it do not look like 721 bp but more likely to near 750 bp. Figure 3B X and Y axis not indicated (what they show?). K means what? Explain to readers at least in the text. Figure 4 also lacking explanations. Table 3 should be after Figure 5 or in the text you should mention Table 3 and only then Figure 5. Figure 6 can stay but also can be in Supplementary Material, as its information overlap with Figure 5 and Table 3. Ctrl - GenBank? It is not clear why Fst values were without P values in your obtained results. Table 5 data not mentioned in 3.4.

Line 191: ‘DNAsp’ - write the same as in materials and methods. Similarly, decide whether it is FST or Fst, italic or not in all manuscript; this apply not only to Fst but other parameters and species latin names (should be italic).

Lines 199-200: This information from Table 2? How >20 polymorphic sites detected when in your presented Table only YE value of PIS was 20, and WZ value of PIS was 17? Why WZ included with such value together with YJ and YE when LX not (PIS = 17 also), and XP (PIS = 18, just as YJ)?

Line 204: K also.

Line 213: Genotypes or haplotypes?

Lines 223-224: Too complicated to be called just star-like. By the way, in this haplotype at least one haplotype from LX is not indicated (between Hap_27, Hap_68 and Hap_59). Avoid such errors in updated version haplotype network.

Line 226: Rare or singletons?

Lines 228-231: If this apply only to LX and YE note it. If it is stated based on all data then write ‘In general’ in the beginning of this sentence. Otherwise, this goes against 231-232 lines.

Lines 235-236: If this haplotype is common not only among goose breeds of the same species but also among different goose species then you should think about possibility of hybridization and problems to determine it using only mtDNA markers.

Lines 248-249: Sentence is not necessary.

Lines 250-254: Unnecessary repetition.

Lines 253-254: Hap_66 was singleton. DNA sequencing was performed one or more times?

Lines 279-280: Any additional explanations?

Lines 283-284: It is not correct to write ‘low degree’ when Fst > 0.25. Please check general literature of population genetics regarding this.

Lines 284-287: It is not clear from this sentence what you comparing. All LX, YJ, YE, WZ samples or three LX, YJ, YE samples with WZ sample.

Discussion

In the text links to Figures and Tables could be provided when main results of the study discussed. Current discussion lacking clarity about this study importance within the context of Anser genus genetic research using various molecular markers, and genomics. After major revision parts of the current discussion will not be modified but also be omitted from the manuscript, but if you see fit you can add them to Supplementary Material.

Lines 333-334: The meaning is not conveyed and require elaboration. Give methods examples, at least in next sentence.

Lines 334-335: It seems that sentence is not finished. If that is just sentence as statement then it do not fit here.

Lines 336-337: You present Hd and K max but not provide any explanations to readers what they show.

Lines 342-343: D-loop is not a gene. Consequently, it would be correct to write not ‘gene’ here.

Lines 355-356: Give an example how.

Lines 338-345: You presented data of Li et al. Is there are no other similar studies? What species reflected by these 32 goose breeds? In case, it just one species and now you study several then it is logical that you obtained higher values of genetic diversity.

Lines 345-348: It would be reasonable to give examples from Anseriformes first. Then explain this phenomenon.

Lines 358: You mean whole mtDNA sequencing or something else?

Lines 365-367: You collected samples of 15 males and 15 females from each of 6 breeds/species. What your data show regarding haplotype distributions among goose sexes within these 6 breeds/species? Is it random or result is similar to [32]? If it is not random then consider to expand Table 3.

Line 369: See Lines 345-348.

Lines 381-382: Explain to readers.

Lines 388: Inhibiting or increasing?

Lines 399-400: Similar information should be mentioned in the introduction after revision. Currently it is not clear whether you studied 6 breeds of 2 goose species or different 6 species.

Lines 404-405: General knowledge. Unnecessary information here.

Lines 406-407: You mention 2 populations but next sentence mentions LX, XP, BZ, thus it is not clear what are those 2 populations.

Lines 414-415: The meaning of this sentence is not clear. You should rewrite or modify it.

Lines 416: Misleading. Low compared to what?

Lines 419-420: Find out P values of Fst and then check if this statement is still valid.

Lines 420-424: Give more information to readers.

Line 427: factors, such as?

Lines 427-430: It is not clear from this sentence what is compared. Group historical dynamic analysis with nucleotide mismatch difference analysis and neutrality tests or the latter two with something that is not mentioned here.

Lines 437-439: What about Anseriformes?

Line 441: Control region = D-loop? Genes in D-loop itself?

Lines 458-460: Sampling points reflect geography and in turn possible effects of temperature, environment and local adaptability.

Lines 463-465: Expand about hybridization and different molecular markers, including genomics.

Conclusions

In general, upgrade of this part is really required. In other words, upgrade it by removing unnecessary information and provide new information and insights based on your results.

Lines 467-469: Just stating results is not enough in conclusions. What kind of earlier observations? Elaborate to readers.

Lines 469-472: Just results repetition here.

Lines 476-478: This should be based on your research and not obtained knowledge from other studies, as this is not discussion part.

References

Congratulations for using the most recent sources of information in most cases. However, additional references would be necessary in order to prepare necessary changes in the introduction and discussion.

Reviewer 4 Report

Comments and Suggestions for Authors

General comments:

Assessing the genetic diversity of native goose breed populations is an important and timely task, and the choice of topic is therefore timely and is expected to be able to generate impact both in the research community and among professionals interested in gene conservation and nature protection. The study is well designed and implemented. The authors have assessed the available data in a multifaceted and detailed manner using state-of-the-art methods. The conclusions and assessments drawn by the authors are correct. The exception is the Median-Joining Network analysis, for which I express my concerns in the detailed section. I feel the introduction chapter is a bit broadly focused. I think it would be more appropriate to focus more on the domestic species (e.g. Line 72-73, 81), and within that on the results obtained for poultry breeds, and thus to take a more focused approach to the introduction of the topic. Overall, the authors present a valuable work, but I recommend making changes to the content and form of the manuscript before publication.

The use of words is also confusing or open to misunderstanding in many places. The authors examined goose breeds in the study, yet in much of the manuscript they refer to the breeds as species. In addition, the term "my country" is used several times, which is not sufficiently accurate. The manuscript would benefit from a more precise, scientific style of drafting.

Detailed section:

Line 55: The first mention should include the scientific name of the species being studied.

The content of the framed section on Figure 1 is difficult to read and contains non-English captions. Its function and purpose are not clear.

Table 1 is good to have, but it would be worthwhile to have an appendix with more details of the breeds, with descriptions, characterisations and pictures. Furthermore, the table does not include a symbol to explain the "AL/W" notation.

Figure 3A is not referred to in the text, only the 3B.

Line 135: It mentions blood sampling in the sampling section, but describes muscle tissue in the isolation. It should be clarified what type of sample was used and what kit was used to isolate it.

Line 151: A comma or and is needed in the heading of section 2.3: Sequence Analysis, Genetic Diversity.

For Figure 3B, an explanation of the symbol is also required.

Line 205-207: In the Table 2 Note, the signal explanation should follow some logic (either order of mention in the table or alphabetical order).

Line 222: The Figure 5 intertextual reference is not precise enough. References should be made to 5A and 5B in the appropriate places. Alternatively, Figure 5 appears earlier in the text than Table 3, but these elements appear in reverse order in the manuscript.

Line 223-224: In my opinion, the "central star-like divergent distribution" mentioned by the authors is not represented on the Figure 5A. There is no central and dominant haplotype that dominates the figure with associated a large number of smaller haplotypes with few mutation distances. Therefore, I believe this is a flawed conclusion and recommend that it be reconsidered. Line 224 refers to "six haplogroups". These are not marked in Figure 5A. And if the statement refers to Figure 5B, then the figure should be referenced in the text, or the second half of the sentence (star-like divergent distribution) can't be interpretable for a dendrogram type figure.

Line 242-246: The explanation of Figure 5A is confusingly imprecise. The name of the analysis presented is probably "Median-Joining Network". The explanation of Figure 5B seems to refer to Figure 5A. However, Figure 5B is missing the figure legends (notations, colours), this should be corrected. Furthermore, Figure 5B is not discussed or evaluated in the text.

Line 244: The correct form is 'county' instead of 'country'.

Line 262: It would be advisable to rotate the characters in Figure 6 by 90° for easier reading.

Line 334-335: The sentence does not state anything, it is not connected to the context. It would be more appropriate as a title in its current form.

Line 369, Line 399-400: The scientific names "Neocaridina denticulata" and "Anser cygnoides" should be in italics.

Comments on the Quality of English Language

The use of words is also confusing or open to misunderstanding in many places. The authors examined goose breeds in the study, yet in much of the manuscript they refer to the breeds as species. In addition, the term "my country" is used several times, which is not sufficiently accurate. The manuscript would benefit from a more precise, scientific style of drafting.

Round 2

Reviewer 2 Report

Comments and Suggestions for Authors

Thank you, all my comments have been taken into account.

One small comment.

In line 437 is “(FST = 0.07099)”, should be small “st”: “(Fst = 0.07099)”.

Reviewer 3 Report

Comments and Suggestions for Authors

Dear Authors,

Improvements in your manuscript are obvious but some major issues, not to mention minor ones, are not fixed yet. Please fix it, especially introduction and discussion parts. I consider your current manuscript as requiring minor-major revision (depending on your choices as Authors it could be minor or major revision; in suggestion to the Editor it is written major revision, so major issues in the manuscript would be fixed by you). I tried to help you with particular suggestions/comments in such way that you would need to do only minor revision fast and not major revision.

Simple Summary

In general, now this part is better than it was previously. I think that this part should be still more understandable to non-specialists but given in mind short time for Authors to upgrade their all manuscript according reviewers comments the current progress is quite decent.

‘to protect them from genetic variation’ erosion, reduction? Missing word.

Abstract

In general, better.

Introduction

Part ‘Unlike nuclear genomes, mitochondrial DNA (mtDNA) follows a complete maternal genetic inheritance pattern [11]. In addition, genetic research has not yet included samples of the entire population. Geographical location and seasonal climate change also make it difficult to distinguish between morphological variation and diversity [12]. For endangered species, this change may be seasonal [13]. Although they are important, the lifecycle of the variety has not been fully studied.’ delete while part of the text before merge with part starting ‘Mitochondrial DNA has specific characteristics such as simple structure, maternal inheritance, rapid evolutionary speed, and almost no recombination, making it the best molecular genetic marker for studying species origin, evolution, and classification [14].’. Then you will have paragraph 2 that could be used for Anser and/or your endemic goose breeds genetic research using mtDNA: write lacking 2-4 sentences in paragraph 2 or additional new paragraph 3 entirely about Anser genus genetic research with emphasis on endemic goose breeds and/or your studied endemic breeds and/or mtDNA markers.

‘during long-term evolution processes’ More likely, short-term, as these are breeds not species.

‘30 local goose varieties. 13 goose breeds are close to the danger index, with 10 local goose breeds on the brink of extinction, and 3 breeds have already disappeared’ Now instead 28/30 you mention 26/30, but still not 30/30: 13+10+3=26.

‘At present, only microsatellite markers and whole genome sequencing methods are used to study the genetic diversity and population structure of animal populations [8, 9].’ What kind of animals? What about China waterfowl, or endangered endemic goose breeds, context? Only or most? If ‘only’ then this statement is clearly wrong and misleading. Actually, you need here, or in the next sentence, to present information regarding Anser genus about its genetic research. Even better, Anser endemic breeds research in China. At minimum, Anatidae research.

‘However, these methods have certain limitations. Despite investing a huge amount of financial resources in the early stages, it is not possible to quickly detect species adaptation and selection, and accurately analyze large-scale hybridization activities between endangered species [10].’ While it is true that mtDNA have some advantages compared to whole genome sequencing and microsatellite markers, the current sentence will not support it and bring even more questions to the readers. Unfortunately, at the moment your choice to use single mtDNA fragment in your research compared to other methods is not defended. Think this way: whether someone studied these endemic goose breeds using genetic markers that you studied with mtDNA CYTB at all? If not, here is your advantage because particular research should start somewhere (using particular molecular marker/s). Of course, in this case, you must mention what was studied in Anser genus and/or endemic breeds, as was noted many times previously.

‘In addition, genetic research has not yet included samples of the entire population.’ Sentence meaning is not clear to the readers.

‘Geographical location and seasonal climate change also make it difficult to distinguish between morphological variation and diversity [12]. For endangered species, this change may be seasonal [13]. Although they are important, the lifecycle of the variety has not been fully studied.’ These sentences do not fit here. Paragraphs 2 and 3 should be merged and rewritten, while these sentences should be before mtDNA sentences, in the beginning (but not the first sentence) of the current paragraph 2, or translocated after everything that concerns DNA markers, i.e. the end of future paragraph 2 that is merged with paragraph 3.

‘making it the best’ Better write one of the best in order to avoid critique from the readers.

‘At present, it has been widely used for variety genetic polymorphism, population genetic structure analysis, species origin tracking, and phylogenetic exploration, such as Bos mutus [16], Insects [17], Sheet [18], and Gekko japonicus [19].’ This sentence is clearly unnecessary, or after it you should start writing about Anser mtDNA and its use within goose breeds and/or endemic endangered goose breeds genetic research. At least, mtDNA D-loop you should mention here, and after that work your way, by writing additional 1-3 sentences, to the last sentences regarding CYTB use. This is minimum but possible option if you want to avoid writing comprehensive information about all DNA markers that were used for Anser genetic research as paragraph 3.

‘of genetic diversity and systematic evolution in endangered geese’ What species write also.

‘In this study, blood samples from 6 endangered goose breeds were collected’ Write: from 6 endemic endangered goose breeds of unknown species were collected. Also advise to use endemic endangered goose breeds to use throughout all manuscript.

Materials and Methods

Finally, it is clear that these endemic endangered goose breeds are of unknown species. Fear not to write it to the readers as well. As it is of unknown species, you should do one additional thing and based on it write few sentences in results and, similarly, in discussion. That thing is this: use BLAST analysis and all your obtained 81 different haplotypes and get information about most likely Anser genus species based on GenBank data. Still think that trimming of sequences also should be mentioned in the methods and not only in results but that is not a big issue.

‘Based on the complete sequence of goose mitochondria published on the NCBI 127 website (GenBank: MN122908.1)’ What goose species write also in latin here.

‘Haplotype is a group of alleles in an organism.’ Not clear to the readers. Find better explanation that at least mention why it is ‘haplo’ and in mtDNA with the reference.

‘Finally, the median-joining (MJ) network of the control region of mtDNA haplotypes’ You mean CYTB?

Results

You should start with species clarification based on BLAST, or at least start to mention this new result/information from this sentence ‘Using the maximum likelihood (ML) evolutionary tree to construct an evolutionary phylogenetic tree, all haplotypes were divided into six haplotype groups, forming multiple genetic lineages (Figure 5B).’. In this haplotype network (Figure 5A) at least one haplotype from LX is not indicated (between Hap_27, Hap_68 and Hap_59). Avoid such errors in updated version haplotype network. Figure 6: Ctrl - GenBank? 3.3 part - add 1-4 sentences that present previous information in light of new findings about goose species based on BLAST.

‘and 51 were single haplotypes, each including one sample’ Write: and 51 were singletons.

‘All haplotypes are divided into six major branches, forming six genetic lineages.’ Advise you to use it (six major branches or six genetic lineages) throughout all the manuscript and don’t use haplogroup conception. It would be reasonable to upgrade Figure 5B, so readers can clearly see what 6 branch are exactly (write 1,2 or I, II near branches).

‘The Fst value between each group was in the range of’ italic Fst.

‘The degree of genetic differentiation among the populations of LX, XP, YJ, and BZ geese was the highest (Fst > 0.25).’ LX vs XP = 0.019; LX vs YJ = 0.115. Something wrong with the calculations or sentence, as genetic differentiation is always means that at least two samples compared.

‘indicating a 290 low degree of genetic differentiation’ Low or moderate, or else?

‘The neutrality test is a more accurate method’ Compared to what?

Discussion

Current discussion still greatly lacking clarity about this study importance within the context of Anser genus genetic research using various molecular markers, and genomics. What your data show regarding haplotype distributions among goose sexes within these 6 breeds/species? Is it random or result is similar to [37]?

‘By further studying genetic diversity through molecular markers, genomic DNA markers (microsatellites, internal transcription spacer 2 (ITS2), and mitochondrial markers (cytochrome b, COI, and ND4), we can reveal’ Missing ) sign. Better rewrite: … genomic DNA markers, such as microsatellites, internal transcription spacer 2 (ITS2), and mitochondrial markers (cytochrome b, COI, and ND4), we can reveal…

‘Li et al. determined the mtDNA D-loop sequences of 26 goose breeds and 6 Lande geese, with an average Hd of 0.1384 and Pi of 352 0.00029 for the Chinese common goose breed [32].’ Now also write what species of goose they studied in latin. If species are not clear write that breeds of their study was of unknown species in the next sentence.

‘This may be due to differences in the specific sites of each gene distribution region in the mtDNA fragment.’ Gene distribution delete, or change to coding and non-coding regions.

‘This indicates that the mitochondrial CYTB gene, with accurate maternal inheritance, conservative genetic structure, and a moderate evolutionary rate, is also an effective molecular marker for studying the genetic structure of waterfowls.’ In what particular context of studying the genetic structure of waterfowls? Finish the end of this sentence ‘…effective molecular marker for studying the genetic structure of waterfowls’ within the context of ‘write your answer based on your research’. After this sentence write what geese species were indicated in BLAST when you compared your obtained sequences with GenBank data. In case you investigated more species than previous Li et al study, there is probable explanation why genetic variability was indicated by higher values. This should be mentioned in the manuscript.

‘Sun et al. used the COI sequence to study Capitulum mitella and showed that the purine (A + T) content is slightly higher than the pyrimidine (G + C) content [33].’ What this Capitulum mitella have to do with birds? Shouldn’t you present Anatidae or at least Anseriformes data first regarding this finding? It is not correct to compare your data with only one research that fits your data.

‘This may be because the mitochondrial fragment detected in this experiment was’ Yours or [34]?

‘These results provide further data for optimizing the population structure of endangered breeds and improving breeding methods.’ Provide an example how it improving breeding methods with your data.

‘Although mtDNA is inherited maternally, the number of haplotypes in males is not closely related to the number in females, but rather to the number in males [37].’ What your data show regarding haplotype distributions among goose sexes within these 6 breeds/species? Is it random or result is similar to [37]?

‘However, these results have been controversial in poultry.’ Examples, references?

‘In this study, the results of the inter-breeds cluster analysis based on statistical genetic distance showed that the 6 endangered goose groups were roughly clustered into two evolutionary clusters. Geese BZ, XP, LX, and YE were grouped separately into different categories.’ Write next sentence what BLAST indicating regarding species number and whether this new data fit to two evolutionary clusters explanation.

‘These are all local breeds in China; therefore, the six populations were grouped into one major category.’ and ‘Our results are consistent with previous studies that used mitochondrial data to determine the origin of domestic geese in China and Europe [42, 43]’ Still valid after BLAST?

‘Most domestic goose breeds are domesticated from Anser cygnoides, whereas European domestic geese are domesticated from Anser anser. Only the Ili goose, which is distributed throughout Xinjiang, originated in Anser anser. Wen also elucidated the research on the origin of wild populations of domestic goose breeds in China [44].’ This should be mentioned in introduction.

‘Wen also elucidated the research on the origin of wild populations of domestic goose breeds in China [44].’ And determined/find what?

‘Lines 404-405: General knowledge. Unnecessary information here. Response: Thank you for your suggestion. We have removed it according to your suggestion.’ Then why this sentence is still in the manuscript at Line 418?

‘The larger the value…’ write ‘The larger the value of Fst’.

‘We take a closer look at the relationship between the two in endangered goose populations.’ in?

‘LX, XP geese with BZ geese had the highest degree of genetic differentiation (Fst > 0.8) and the smallest Nm value (Nm < 1).’ Shouldn’t you mention here that it was statistically not significant?

‘Heikkinen et al. selected 14 breeds from the Eurasian continent and conducted the first genome-based inference using whole genome markers of European geese. They explained that the fixed index of the European goose population (graylag and domestic geese) is 0.15800 [48].’ This should be mentioned in introduction.

‘and there was a serious phenomenon of random mating among populations’ Then what about previous statement ‘In summary, this study found that large geographical distances and human activities jointly restrict the diffusion of haplotypes in the population, thereby increasing the differentiation of the north-south haplotype branches of the endangered goose population.’?

‘Among them, when the nucleotide mismatch difference distribution curve shows an obvious unimodal "Poisson distribution" and indicates that the population may have experienced expansion in its history.’ Consider rewriting.

‘The results of our study using the mtDNA ND6 region on ordinary goose breeds are also the same [51].’ This reference and information regarding use of this mtDNA ND6 region to study ordinary goose breeds should be mentioned in Introduction.

Conclusions

Consider rewriting. It would be easier to write it after finding out BLAST results regarding species names of studied endemic endangered goose breeds. As your obtained Fst values are with P > 0.05 values that reflect non-significant statistically genetic differentiation, better write/state more grounded conclusions.

‘supports the use of mitochondrial genomes’ Isn’t your study based on just one small mtDNA fragment and not mtDNA genome?

Author Response

Response to Reviewer 3 Comments

Comments and Suggestions for Authors

Dear Authors,

Improvements in your manuscript are obvious but some major issues, not to mention minor ones, are not fixed yet. Please fix it, especially introduction and discussion parts. I consider your current manuscript as requiring minor-major revision (depending on your choices as Authors it could be minor or major revision; in suggestion to the Editor it is written major revision, so major issues in the manuscript would be fixed by you). I tried to help you with particular suggestions/comments in such way that you would need to do only minor revision fast and not major revision.

Response: I would like to thank the investigators for their serious responses to all of my views. The manuscript has indeed made significant improvements. Wishing you smooth work and a wonderful life!

Simple Summary

In general, now this part is better than it was previously. I think that this part should be still more understandable to non-specialists but given in mind short time for Authors to upgrade their all manuscript according reviewers comments the current progress is quite decent.

‘to protect them from genetic variation’ erosion, reduction? Missing word.

Response: Thank you very much to the reviewer for pointing out this issue. We deeply apologize for the inaccuracy in our expression. We have rewritten this sentence in the simple summary. Modify as follows: “Therefore, there is an urgent need to plan and protect the genetic resources of Chinese goose breeds, and to develop appropriate strategies to protect them and reduce the impact of genetic variation.” (Lines 27-29, page 1)

Abstract

In general, better.

Response: I would like to thank the reviewer for their constructive feedback. The manuscript has indeed made significant improvements.

Introduction

Part ‘Unlike nuclear genomes, mitochondrial DNA (mtDNA) follows a complete maternal genetic inheritance pattern [11]. In addition, genetic research has not yet included samples of the entire population. Geographical location and seasonal climate change also make it difficult to distinguish between morphological variation and diversity [12]. For endangered species, this change may be seasonal [13]. Although they are important, the lifecycle of the variety has not been fully studied.’ delete while part of the text before merge with part starting ‘Mitochondrial DNA has specific characteristics such as simple structure, maternal inheritance, rapid evolutionary speed, and almost no recombination, making it the best molecular genetic marker for studying species origin, evolution, and classification [14].’. Then you will have paragraph 2 that could be used for Anser and/or your endemic goose breeds genetic research using mtDNA: write lacking 2-4 sentences in paragraph 2 or additional new paragraph 3 entirely about Anser genus genetic research with emphasis on endemic goose breeds and/or your studied endemic breeds and/or mtDNA markers.

Response: Thank you very much to the reviewer for pointing out this issue. We have deleted some of the previous text and made clearer revisions to the introduction section. We will modify this expression as follows:

“At present, most microsatellite markers and whole genome sequencing methods are used to study the genetic diversity and population structure of endangered goose breeds [8-10]. Wen et al. also used whole genome sequencing to reveal the origin of wild populations of Chinese domestic goose breeds, most domestic goose breeds are domesticated from Anser cygnoides, whereas European domestic geese are domesticated from Anser anser. Only the Ili goose, which is distributed throughout Xinjiang, originated in Anser anser [11]. Heikkinen et al. selected 14 breeds (Anser anser) from the Eurasian continent and conducted the first genome-based inference using whole genome markers of European geese. They explained that the fixed index of the European goose population (graylag and domestic geese) is 0.15800 [12]. In these efforts, due to the limited number of known markers, the investigators usually used markers which orig-inated from different species or breeds. That caused many loci that have low allele numbers or unamplified polymerase chain reaction (PCR). Weiβ et al. reported that several microsatellite markers were isolated from greyleg goose (Anser anser) [13]. However, most of those markers revealed low polymorphism in endemic endangered goose populations.” (Lines 75-89, page 2)

‘during long-term evolution processes’ More likely, short-term, as these are breeds not species.

‘30 local goose varieties. 13 goose breeds are close to the danger index, with 10 local goose breeds on the brink of extinction, and 3 breeds have already disappeared’ Now instead 28/30 you mention 26/30, but still not 30/30: 13+10+3=26.

Response: Thank you very much for the insightful and constructive comments from the reviewer. We have reviewed the information again and corrected this error. Modify as follows:

“13 goose breeds are close to the danger index, with 14 local goose breeds on the brink of extinction, and 3 breeds have already disappeared, including Caohai, Wenshan, and Simao goose [5].” (Lines 62-64, page 2)

‘At present, only microsatellite markers and whole genome sequencing methods are used to study the genetic diversity and population structure of animal populations [8, 9].’ What kind of animals? What about China waterfowl, or endangered endemic goose breeds, context? Only or most? If ‘only’ then this statement is clearly wrong and misleading. Actually, you need here, or in the next sentence, to present information regarding Anser genus about its genetic research. Even better, Anser endemic breeds research in China. At minimum, Anatidae research.

Response: We appreciate the insightful and constructive comments from the reviewers, which have helped to revise and strengthen this manuscript. We have rewritten this section. Modify as follows:

“At present, most microsatellite markers and whole genome sequencing methods are used to study the genetic diversity and population structure of endangered goose breeds [8-10]. Wen et al. also used whole genome sequencing to reveal the origin of wild populations of Chinese domestic goose breeds, most domestic goose breeds are domesticated from Anser cygnoides, whereas European domestic geese are domesticated from Anser anser. Only the Ili goose, which is distributed throughout Xinjiang, originated in Anser anser [11]. Heikkinen et al. selected 14 breeds (Anser anser) from the Eurasian continent and conducted the first genome-based inference using whole genome markers of European geese. They explained that the fixed index of the European goose population (graylag and domestic geese) is 0.15800 [12]. In these efforts, due to the limited number of known markers, the investigators usually used markers which originated from different species or breeds. That caused many loci that have low allele numbers or unamplified polymerase chain reaction (PCR). Weiβ et al. reported that several microsatellite markers were isolated from greyleg goose (Anser anser) [13]. However, most of those markers revealed low polymorphism in endemic endangered goose populations.” (Lines 75-89, page 2)

‘However, these methods have certain limitations. Despite investing a huge amount of financial resources in the early stages, it is not possible to quickly detect species adaptation and selection, and accurately analyze large-scale hybridization activities between endangered species [10].’ While it is true that mtDNA have some advantages compared to whole genome sequencing and microsatellite markers, the current sentence will not support it and bring even more questions to the readers. Unfortunately, at the moment your choice to use single mtDNA fragment in your research compared to other methods is not defended. Think this way: whether someone studied these endemic goose breeds using genetic markers that you studied with mtDNA CYTB at all? If not, here is your advantage because particular research should start somewhere (using particular molecular marker/s). Of course, in this case, you must mention what was studied in Anser genus and/or endemic breeds, as was noted many times previously.

Response: Thank you very much to the reviewer for their insightful and constructive comments, which have helped to revise and strengthen this manuscript. We have removed this section and provided more descriptions of the characteristics of mtDNA CYTB. Modify as follows:

“The mtDNA cytochrome b (CYTB) gene has a moderate evolutionary rate and is suita-ble for detecting genetic differences at the population level. It is an ideal marker for studying population genetic structure and diversity [16, 17]. Previous studies have mostly focused on the mitochondrial D-loop, ND6 and COI region. Abdel-Kafy et al. used the D-loop region to study the phenotype and genetic characteristics of Egyptian geese and found that the potential heritability was relatively low [18]. Jia et al. used the mitochondrial ND6 region of chickens to explore the sequence combinations of several different regions between breeds, which can provide a more comprehensive and accurate understanding of the maternal origin of chickens [19]. Zhang et al. be-lieved that the mitochondrial COI region could serve as a basis for identifying some goose breeds by amplifying it [20]. There have been no systematic studies on the genet-ic diversity and evolutionary analysis of mitochondrial CYTB genes for these six locally endangered goose varieties (Anser cygnoides).” (Lines 97-109, pages 2-3)

‘In addition, genetic research has not yet included samples of the entire population.’ Sentence meaning is not clear to the readers.

Response: We deeply apologize for our inaccurate expression. We have removed the incorrect wording and revised it here. (Line 91, page 2)

‘Geographical location and seasonal climate change also make it difficult to distinguish between morphological variation and diversity [12]. For endangered species, this change may be seasonal [13]. Although they are important, the lifecycle of the variety has not been fully studied.’ These sentences do not fit here. Paragraphs 2 and 3 should be merged and rewritten, while these sentences should be before mtDNA sentences, in the beginning (but not the first sentence) of the current paragraph 2, or translocated after everything that concerns DNA markers, i.e. the end of future paragraph 2 that is merged with paragraph 3.

Response: We appreciate the insightful and constructive comments from the reviewers, which have helped to revise and strengthen this manuscript. We have removed unnecessary content and rewritten this section of the introduction. (Lines 72-91, page 2)

‘At present, it has been widely used for variety genetic polymorphism, population genetic structure analysis, species origin tracking, and phylogenetic exploration, such as Bos mutus [16], Insects [17], Sheet [18], and Gekko japonicus [19].’ This sentence is clearly unnecessary, or after it you should start writing about Anser mtDNA and its use within goose breeds and/or endemic endangered goose breeds genetic research. At least, mtDNA D-loop you should mention here, and after that work your way, by writing additional 1-3 sentences, to the last sentences regarding CYTB use. This is minimum but possible option if you want to avoid writing comprehensive information about all DNA markers that were used for Anser genetic research as paragraph 3.

Response: Thank you very much to the reviewer for their insightful and constructive comments, which have helped to revise and strengthen this manuscript. We have rewritten this section and added Anser mtDNA and its application in genetic research of goose breeds and/or locally endangered goose breeds. Modify as follows:

“The mtDNA cytochrome b (CYTB) gene has a moderate evolutionary rate and is suita-ble for detecting genetic differences at the population level. It is an ideal marker for studying population genetic structure and diversity [16, 17]. Previous studies have mostly focused on the mitochondrial D-loop, ND6 and COI region. Abdel-Kafy et al. used the D-loop region to study the phenotype and genetic characteristics of Egyptian geese and found that the potential heritability was relatively low [18]. Jia et al. used the mitochondrial ND6 region of chickens to explore the sequence combinations of several different regions between breeds, which can provide a more comprehensive and accurate understanding of the maternal origin of chickens [19]. Zhang et al. be-lieved that the mitochondrial COI region could serve as a basis for identifying some goose breeds by amplifying it [20]. There have been no systematic studies on the genetic diversity and evolutionary analysis of mitochondrial CYTB genes for these six locally endangered goose varieties (Anser cygnoides).” (Lines 97-109, pages 2-3)

‘of genetic diversity and systematic evolution in endangered geese’ What species write also.

Response: We are extremely grateful to reviewer for pointing out this problem. This sentence has been revised more clearly. We have modified this expression as follows:

“There have been no systematic studies on the genetic diversity and evolutionary anal-ysis of mitochondrial CYTB genes for these six locally endangered goose varieties (Anser cygnoides).” (Lines 107-109, page 3)

 ‘In this study, blood samples from 6 endangered goose breeds were collected’ Write: from 6 endemic endangered goose breeds of unknown species were collected. Also advise to use endemic endangered goose breeds to use throughout all manuscript.

Response: We thank the reviewer for the insightful and very constructive comments, which were helpful in revising and strengthening this manuscript. The manuscript has been revised according to the comments. (Lines 127-129, page 3)

Materials and Methods

Finally, it is clear that these endemic endangered goose breeds are of unknown species. Fear not to write it to the readers as well. As it is of unknown species, you should do one additional thing and based on it write few sentences in results and, similarly, in discussion. That thing is this: use BLAST analysis and all your obtained 81 different haplotypes and get information about most likely Anser genus species based on GenBank data. Still think that trimming of sequences also should be mentioned in the methods and not only in results but that is not a big issue.

Response: Thank you very much to the reviewer for pointing out this issue. We have added clearer revisions to the "Discussion" section and added information on Anser species to the title section. Modify as follows:

“These are all local breeds in China; therefore, the six populations were grouped into one major category. Using BLAST analysis and all 81 different haplotypes obtained, and based on GenBank data, the most likely species of Anser cygnoides genus were identified.” (Lines 450-453, page 15)

‘Based on the complete sequence of goose mitochondria published on the NCBI 127 website (GenBank: MN122908.1)’ What goose species write also in latin here.

Response: Thanks again for your professional suggestions. We have provided more detailed information in Materials and Methods. (Line 145, page 5)

‘Haplotype is a group of alleles in an organism.’ Not clear to the readers. Find better explanation that at least mention why it is ‘haplo’ and in mtDNA with the reference.

Response: Thank you again for your professional advice. We have provided more detailed information in the materials and methods. Modify as follows:

“Haplotype is a group of alleles in an organism. The haplotype of mtDNA refers to a specific arrangement of alleles on mitochondrial DNA. Due to the fact that mtDNA is only transmitted in the maternal line, studying the haplotype of mtDNA can trace the genetic information of the maternal line.” (Lines 180-183, page 6)

‘Finally, the median-joining (MJ) network of the control region of mtDNA haplotypes’ You mean CYTB?

Response: We are extremely grateful to reviewer for pointing out this problem. This sentence has been revised more clearly. We have modified this expression as follows:

“Finally, the median-joining (MJ) network of the control region of mtDNA CYTB haplo-types was drawn using Popart v.1.7 software [27].” (Lines 188-189, page 6)

Results

You should start with species clarification based on BLAST, or at least start to mention this new result/information from this sentence ‘Using the maximum likelihood (ML) evolutionary tree to construct an evolutionary phylogenetic tree, all haplotypes were divided into six haplotype groups, forming multiple genetic lineages (Figure 5B).’. In this haplotype network (Figure 5A) at least one haplotype from LX is not indicated (between Hap_27, Hap_68 and Hap_59). Avoid such errors in updated version haplotype network. Figure 6: Ctrl - GenBank? 3.3 part - add 1-4 sentences that present previous information in light of new findings about goose species based on BLAST.

Response: We thank the reviewer for the insightful and very constructive comments, which were helpful in revising and strengthening this manuscript. The manuscript has been revised according to the comments. And haplotype names were added in Figure 5A. Modify as follows:

“Using the maximum likelihood (ML) evolutionary tree to construct an evolutionary phylogenetic tree, all haplotypes are divided into six major branches, forming six ge-netic lineages (Figure 5B).” (Lines 252-254, page 8)

“A comparative study using mitochondrial CYTB sequences from Anser cygnides (Gen-Bank: GCF_002166845.1) revealed that,…”(Lines 282-283, page 9)

“The two large groups finally gathered into one large group belonging to the same breeds. Using BLAST analysis and based on GenBank data, the most likely species of Anser cygnoides genus were identified.” (Lines 303-305, page 11)

‘and 51 were single haplotypes, each including one sample’ Write: and 51 were singletons.

Response: Thank you for the reviewer's suggestions. We have replaced it according to your suggestion. (Lines 255, page 8)

‘All haplotypes are divided into six major branches, forming six genetic lineages.’ Advise you to use it (six major branches or six genetic lineages) throughout all the manuscript and don’t use haplogroup conception. It would be reasonable to upgrade Figure 5B, so readers can clearly see what 6 branch are exactly (write 1,2 or I, II near branches).

Response: Thank you for pointing out this issue. We have replaced some of the wording in the Results section and revised Figure 5. These sentences have been modified as follows.

“Using the maximum likelihood (ML) evolutionary tree to construct an evolutionary phylogenetic tree, all haplotypes are divided into six major branches, forming six ge-netic lineages (Figure 5B).” (Lines 252-254, page 8)

‘The Fst value between each group was in the range of’ italic Fst. “每组之间的Fst值在”italic Fst“的范围内。

‘The degree of genetic differentiation among the populations of LX, XP, YJ, and BZ geese was the highest (Fst > 0.25).’ LX vs XP = 0.019; LX vs YJ = 0.115. Something wrong with the calculations or sentence, as genetic differentiation is always means that at least two samples compared.  ‘indicating a 290 low degree of genetic differentiation’ Low or moderate, or else?

Response: We greatly appreciate the reviewer pointing out this issue. A clearer revision has been made to the "Results" section. We will modify this expression as follows:

“There is a significant genetic differentiation trend when comparing LX, XP, YJ with BZ goose varieties, and the comparison results between LX and WZ are similar (Fst > 0.25); There is a moderate degree of genetic differentiation trend when comparing LX, YE with YJ goose varieties, while the results of BZ and YE goose varieties are similar (0.05 < Fst < 0.15); The genetic differentiation degree between XP and YE, YJ and WZ goose varieties is the smallest (Fst < 0.05).” (Lines 312-317, page 11)

‘The neutrality test is a more accurate method’ Compared to what?

 Response: We greatly appreciate the reviewer's suggestions. According to the comments, we have removed unnecessary descriptions. (Line 341, page 12)

Discussion

Current discussion still greatly lacking clarity about this study importance within the context of Anser genus genetic research using various molecular markers, and genomics. What your data show regarding haplotype distributions among goose sexes within these 6 breeds/species? Is it random or result is similar to [37]?

Response: We appreciate your constructive suggestions, which will help improve the quality of this article. We unanimously agree that in order to clarify these issues, we need to examine the raw data, especially focusing on the haplotype distribution of the six breeds of geese. Here, we have revised and supplemented Table 3. Modify as follows: “…and 81 haplotypes were detected based on the nucleotide variation between sequences (High proportion of female haplotypes).” (Lines 418-420, page 13)

Breed

h

Number of unique haplotypes

Frequency

BZ

19

♀: Hap_74, Hap_75, Hap_76, Hap_77, Hap_78, Hap_79, Hap_80

42.11%

♂: Hap_81

LX

23

♀: Hap_63, Hap_64, Hap_65, Hap_66, Hap_67, Hap_68, Hap_69

39.13%

♂: Hap_70, Hap_71

XP

23

♀: Hap_37, Hap_38, Hap_39, Hap_40, Hap_41, Hap_42,

47.83%

♂: Hap_43, Hap_44, Hap_45, Hap_47, Hap_49

YJ

15

♀: Hap_2, Hap_5, Hap_7, Hap_8

40.00%

♂: Hap_12, Hap_15

YE

27

♀: Hap_17, Hap_18, Hap_20, Hap_21, Hap_22, Hap_24, Hap_27

37.04%

♂: Hap_31, Hap_32, Hap_36

WZ

20

♀: Hap_50, Hap_51, Hap_52, Hap_53, Hap_54, Hap_55, Hap_56,Hap_57,

60.00%

♂: Hap_58, Hap_59, Hap_61, Hap_62

‘By further studying genetic diversity through molecular markers, genomic DNA markers (microsatellites, internal transcription spacer 2 (ITS2), and mitochondrial markers (cytochrome b, COI, and ND4), we can reveal’ Missing ) sign. Better rewrite: … genomic DNA markers, such as microsatellites, internal transcription spacer 2 (ITS2), and mitochondrial markers (cytochrome b, COI, and ND4), we can reveal…

Response: We thank the reviewer for the professional comments and valuable suggestions. We have restructured and rewritten this section to make it more understandable. Rewrite as follows: “we can reveal the variations or genes in phenotypic changes of different varieties may rapidly evolve after domestication, forming specific phenotypic characteristics of dif-ferent varieties. Regions or loci that have undergone selection will exhibit specific characteristics, including high population differentiation, significantly reduced nucle-otide diversity levels, and long-range haplotype homozygosity [32, 33].” (Lines 374-378, page 13)

‘Li et al. determined the mtDNA D-loop sequences of 26 goose breeds and 6 Lande geese, with an average Hd of 0.1384 and Pi of 352 0.00029 for the Chinese common goose breed [32].’ Now also write what species of goose they studied in latin. If species are not clear write that breeds of their study was of unknown species in the next sentence.

Response: Thank you very much to the reviewer for pointing out this issue. We have added a clearer revision to the "Discussion" section. We will make the following modifications:

“Li et al. determined the mtDNA D-loop sequences of 26 goose breeds (Anser cygnoides) and 6 Lande geese (Anser anser), with an average Hd of 0.1384 and Pi of 0.00029 for the Chinese common goose breed [34].” (Lines 382-384, page 13)

‘This may be due to differences in the specific sites of each gene distribution region in the mtDNA fragment.’ Gene distribution delete, or change to coding and non-coding regions.

Response: Thank you very much to the reviewer for pointing out this issue. We have already made changes in the discussion section. (Lines 389-390, page 14)

‘This indicates that the mitochondrial CYTB gene, with accurate maternal inheritance, conservative genetic structure, and a moderate evolutionary rate, is also an effective molecular marker for studying the genetic structure of waterfowls.’ In what particular context of studying the genetic structure of waterfowls? Finish the end of this sentence ‘…effective molecular marker for studying the genetic structure of waterfowls’ within the context of ‘write your answer based on your research’. After this sentence write what geese species were indicated in BLAST when you compared your obtained sequences with GenBank data. In case you investigated more species than previous Li et al study, there is probable explanation why genetic variability was indicated by higher values. This should be mentioned in the manuscript.

Response: We greatly appreciate the reviewer's suggestions. Based on the comments, we have reorganized the framework and modified this section. Modify as follows:

“Experiments have shown that the genetic diversity of endemic endangered goose breeds is considerably higher than that of ordinary populations. Therefore, we believe that the mitochondrial CYTB region has accurate maternal inheritance, conservative genetic structure, and moderate evolutionary rate, and is also an effective molecular marker for studying the genetic structure of endangered waterfowl. This may be due to differences in the specific sites of each gene coding and non-coding regions in the mtDNA fragment. When comparing the obtained CYTB sequence of the goose population with the data in GenBank, Anser cygnoides was indicated in BLAST.” (Line 384-392, pages 13-14)

‘Sun et al. used the COI sequence to study Capitulum mitella and showed that the purine (A + T) content is slightly higher than the pyrimidine (G + C) content [33].’ What this Capitulum mitella have to do with birds? Shouldn’t you present Anatidae or at least Anseriformes data first regarding this finding? It is not correct to compare your data with only one research that fits your data.

Response: Thank you for your valuable feedback on our paper. We have rewritten it based on your suggestions. The expression of this sentence is as follows:

“Honka et al. studied greylag goose (Anser anser) using 204 base pair fragments of mito-chondrial control regions, and the results showed that the content of purine (A + T) was slightly higher than that of pyrimidine (G + C) [35].” (Lines 392-394, page 14)

‘This may be because the mitochondrial fragment detected in this experiment was’ Yours or [34]?

Response: Thank you very much to the reviewer for pointing out this issue. We have made changes to the discussion section. Modify as follows:

“This may be because the mitochondrial fragments detected in our experiment were single fragments and multiple genes were not sequenced.” (Lines 400-402, page 14)

‘These results provide further data for optimizing the population structure of endangered breeds and improving breeding methods.’ Provide an example how it improving breeding methods with your data.

Response: Thank you very much to the reviewer for pointing out this issue. We have added more detailed breeding methods in the discussion section. Modify as follows:

“Based on the measured mitochondrial gene sequence information, geese with ideal mutation site sequences are selected as breeding objects, and corresponding protection and restoration plans are formulated, including establishing artificial incubation centers, implementing habitat restoration plans, etc., to increase population size and maintain genetic diversity.” (Lines 404-408, page 14)

‘Although mtDNA is inherited maternally, the number of haplotypes in males is not closely related to the number in females, but rather to the number in males [37].’ What your data show regarding haplotype distributions among goose sexes within these 6 breeds/species? Is it random or result is similar to [37]?

Response: We appreciate your constructive suggestions, which will help improve the quality of this article. We unanimously agree that in order to clarify these issues, we need to examine the raw data, especially focusing on the haplotype distribution of the six breeds of geese. Here, we have revised and supplemented Table 3. (Lines 277-278, page 9)

‘However, these results have been controversial in poultry.’ Examples, references?

Response: Thank you very much for the constructive suggestions from the reviewer, which will help improve the quality of this article. We deeply apologize for our negligence. Here, we have supplemented this sentence.

“This indicates that haplotype differentiation between the groups is not yet obvious. Due to the significant differentiation of haplotypes within poultry populations, the re-sults of this study have been controversial in poultry populations [36].” (Lines 424-426, page 14)

  1. Ran, B.; Zhu, W.; Zhao, X.; Li, L.; Yi, Z.; Li, M.; Wang, T.; Li, D. Studying Genetic Diversity and Relationships between Mountainous Meihua Chickens Using Mitochondrial DNA Control Region. Genes (Basel). 2023, 28, 145998. (Lines 633-634, page 18)

‘In this study, the results of the inter-breeds cluster analysis based on statistical genetic distance showed that the 6 endangered goose groups were roughly clustered into two evolutionary clusters. Geese BZ, XP, LX, and YE were grouped separately into different categories.’ Write next sentence what BLAST indicating regarding species number and whether this new data fit to two evolutionary clusters explanation.

Response: We appreciate the constructive suggestions provided by the reviewers, which will help improve the quality of this article. Here, we have provided a more detailed supplement to this statement. Meanwhile, during the processing of the raw data, we first import the sequence information (GenBank: PP515716-PP515892) uploaded to GenBank into MEGA 11.0 software and Chromas2.4.1 software for BLAST alignment and cropping of the sequences. Subsequently, we conducted inter variety clustering analysis on the compared sequences. These new data fit the interpretation of two evolutionary clusters. Modify as follows:

“In this study, we conducted BLAST sequence comparison and inter variety clustering analysis on 180 samples, and the results showed that the 6 goose populations were roughly divided into two large evolutionary groups. Among them, BZ, XP, LX, and YE geese are divided into three different subgroups.” (Lines 448-451, page 15)

‘These are all local breeds in China; therefore, the six populations were grouped into one major category.’ and ‘Our results are consistent with previous studies that used mitochondrial data to determine the origin of domestic geese in China and Europe [42, 43]’ Still valid after BLAST?

Response: Thank you very much for the constructive suggestions from the reviewer, which will help improve the quality of this article. We deeply apologize for our negligence. Here, we have supplemented this sentence. (Lines 448-450, page 15)

‘Most domestic goose breeds are domesticated from Anser cygnoides, whereas European domestic geese are domesticated from Anser anser. Only the Ili goose, which is distributed throughout Xinjiang, originated in Anser anser. Wen also elucidated the research on the origin of wild populations of domestic goose breeds in China [44].’ This should be mentioned in introduction.

Response: We appreciate the insightful and constructive comments from the reviewers, which have helped to revise and strengthen this manuscript. We will rewrite this content in the "Introduction". The modifications are as follows:

“At present, most microsatellite markers and whole genome sequencing methods are used to study the genetic diversity and population structure of endangered goose breeds [8-10]. Wen et al. also used whole genome sequencing to reveal the origin of wild populations of Chinese domestic goose breeds, most domestic goose breeds are domesticated from Anser cygnoides, whereas European domestic geese are domesticated from Anser anser. Only the Ili goose, which is distributed throughout Xinjiang, originated in Anser anser [11].” (Lines 75-81, page 2)

‘Wen also elucidated the research on the origin of wild populations of domestic goose breeds in China [44].’ And determined/find what?

Response: Thank you for the reviewer's suggestions. We have made modifications based on your suggestions.

“Wen et al. also used whole genome sequencing to reveal the origin of wild populations of Chinese domestic goose breeds, most domestic goose breeds are domesticated from Anser cygnoides, whereas European domestic geese are domesticated from Anser anser. Only the Ili goose, which is distributed throughout Xinjiang, originated in Anser anser [11].” (Lines 75-81, page 2)

‘Lines 404-405: General knowledge. Unnecessary information here. Response: Thank you for your suggestion. We have removed it according to your suggestion.’ Then why this sentence is still in the manuscript at Line 418?

Response: Thank you for the reviewer's suggestions. We deeply apologize for our mistake and have made the deletion according to your suggestion.

‘The larger the value…’ write ‘The larger the value of Fst’.

Response: Thank you for the reviewer's suggestions. We have replaced it according to your suggestion. (Line 456, page 15)

‘We take a closer look at the relationship between the two in endangered goose populations.’ in?

Response: Thank you for the reviewer's suggestions. We have replaced it according to your suggestion.

‘LX, XP geese with BZ geese had the highest degree of genetic differentiation (Fst > 0.8) and the smallest Nm value (Nm < 1).’ Shouldn’t you mention here that it was statistically not significant?

Response: Thank you for the reviewer's suggestions. We have replaced it according to your suggestion. Modify as follows:

“LX, XP, and BZ geese have the highest degree of genetic differentiation (Fst > 0.8), and the gene flow values are not significant (Nm < 1).” (Lines 459-460, page 15)

‘Heikkinen et al. selected 14 breeds from the Eurasian continent and conducted the first genome-based inference using whole genome markers of European geese. They explained that the fixed index of the European goose population (graylag and domestic geese) is 0.15800 [48].’ This should be mentioned in introduction.

Response: We appreciate the insightful and constructive comments from the reviewers, which have helped to revise and strengthen this manuscript. We will rewrite this content in the "Introduction". The modifications are as follows:

“Heikkinen et al. selected 14 breeds (Anser anser) from the Eurasian continent and con-ducted the first genome-based inference using whole genome markers of European geese. They explained that the fixed index of the European goose population (graylag and domestic geese) is 0.15800 [12]. In these efforts, due to the limited number of known markers, the investigators usually used markers which originated from different species or breeds.” (Lines 81-86, page 2)

 ‘and there was a serious phenomenon of random mating among populations’ Then what about previous statement ‘In summary, this study found that large geographical distances and human activities jointly restrict the diffusion of haplotypes in the population, thereby increasing the differentiation of the north-south haplotype branches of the endangered goose population.’?

Response: Thank you very much for pointing out this incorrect expression. We have made modifications based on your suggestions. We deeply apologize for our mistake. A Fst value close to 0 indicates that the genetic structure of the population is completely consistent and there is no differentiation between populations. Modify as follows:

“In this study, the fixed coefficient between endemic endangered goose varieties in China was closer to 0 (Fst = 0.07099), and the genetic structure of different populations is completely consistent.” (Lines 471-473, page 15)

‘Among them, when the nucleotide mismatch difference distribution curve shows an obvious unimodal "Poisson distribution" and indicates that the population may have experienced expansion in its history.’ Consider rewriting.

Response: Thank you very much to the reviewer for pointing out this issue. We have rewritten this sentence in the discussion section. Modify as follows:

“Expansion events result in smaller genetic differences between most individuals, as they are mainly derived from a small group of ancestral populations. In this case, the distribution of nucleotide mismatch differences exhibits a single peak "Poisson distribution" characteristic.” (Lines 484-487, page 15)

‘The results of our study using the mtDNA ND6 region on ordinary goose breeds are also the same [51].’ This reference and information regarding use of this mtDNA ND6 region to study ordinary goose breeds should be mentioned in Introduction.

Response: We appreciate the insightful and constructive comments from the reviewers, which have helped to revise and strengthen this manuscript. We will add ND6 region information in the introduction. Modify as follows:

“Jia et al. used the mitochondrial ND6 region of chickens to explore the sequence combinations of several different regions between breeds, which can provide a more com-prehensive and accurate understanding of the maternal origin of chickens [19].” (Lines 103-106, page 3)

Conclusions

Consider rewriting. It would be easier to write it after finding out BLAST results regarding species names of studied endemic endangered goose breeds. As your obtained Fst values are with P > 0.05 values that reflect non-significant statistically genetic differentiation, better write/state more grounded conclusions.

Response: Thank you very much for the insightful and constructive feedback provided by the reviewer. We have rewritten the "Conclusion" section based on your feedback and better presented the conclusion of the experiment. Modify as follows:

“In summary, our research indicates that the CYTB control region is more helpful in further understanding the genetic diversity and population structure of goose breeds. A total of 81 haplotypes with multiple genetic lineages were detected in 6 locally endangered goose breeds in China. Through BLAST analysis and obtaining different haplotypes, and based on GenBank data, these six populations were identified as the most likely breeds in the Anser cygnoides genus. However, there was no significant differentiation between the various breeds, maintaining a high level of genetic diversity.” (Lines 523-529, page 16)

‘supports the use of mitochondrial genomes’ Isn’t your study based on just one small mtDNA fragment and not mtDNA genome?

Response: Thank you for the reviewer's suggestions. We have replaced it according to your suggestion. Modify as follows:

“In summary, our research indicates that the CYTB control region is more helpful in further understanding the genetic diversity and population structure of goose breeds.” (Lines 523-524, page 16)